# DO DEEP NEURAL NETWORK SOLUTIONS FORM A STAR DOMAIN?

**Ankit Sonthalia**[1][*]    **Alexander Rubinstein**[1]    **Ehsan Abbasnejad**[2,3]    **Seong Joon Oh**[1]
[1]Tübingen AI Center, Universität Tübingen
[2]University of Adelaide
[3]Monash University

## ABSTRACT

It has recently been conjectured that neural network solution sets reachable via stochastic gradient descent (SGD) are convex, considering permutation invariances (Entezari et al., 2021). This means that a linear path can connect two independent solutions with low loss, given the weights of one of the models are appropriately permuted. However, current methods to test this theory often require very wide networks to succeed (Ainsworth et al., 2022; Benzing et al., 2022). In this work, we conjecture that more generally, the SGD solution set is a *star domain* that contains a *star model* that is linearly connected to all the other solutions via paths with low loss values, modulo permutations. We propose the *Starlight* algorithm that finds a star model of a given learning task. We validate our claim by showing that this star model is linearly connected with other independently found solutions. As an additional benefit of our study, we demonstrate better uncertainty estimates on Bayesian Model Averaging over the obtained star domain. Further, we demonstrate star models as potential substitutes for model ensembles. Our code is available at https://github.com/aktsonthalia/starlight.

## 1 INTRODUCTION

The learning problem for a neural network is inherently characterized by a non-convex loss landscape, leading to multiple possible solutions rather than a singular one. Efforts to comprehend this landscape and the set of solutions have been ongoing.

A significant early discovery in this area (Garipov et al., 2018) demonstrated that almost any two independent solutions could be connected through a simple low-loss curve. While this finding highlighted the vastness of the solution set, other research has focused on its complexity. For instance, permutation symmetries allow neuron positions in different layers to be jointly swapped without changing the function represented by the neural network (Brea et al., 2019; Singh & Jaggi, 2020; Ainsworth et al., 2022; Guerrero Peña et al., 2023). Entezari et al. (2021) proposed that when accounting for these symmetries, the solution set found by stochastic gradient descent (SGD) essentially becomes convex, *i.e.,* any pair of independent solutions can be connected through a low-loss line segment after an appropriate permutation is applied to one of the models. Notably, Sharma et al. (2024) investigate the stronger property of *simultaneous* linear connectivity, wherein permuting a given model linearly connects it to *several* other models. However, recent works like Ainsworth et al. (2022) study convexity in the context of the formulation in Entezari et al. (2021). Our work therefore refers to their conjecture as the "convexity conjecture" (Conjecture 1) while acknowledging that other, stronger forms of convexity can be formulated.

The convexity conjecture has faced challenges. Subsequent studies (Juneja et al., 2022; Benzing et al., 2022; Ainsworth et al., 2022; Altintas et al., 2023; Guerrero Peña et al., 2023) revealed that even after the application of permutation-finding algorithms, two distinct solutions in the parameter space might still be separated by a high loss barrier (Frankle et al., 2020; Entezari et al., 2021) upon performing linear interpolation. These studies attribute this discrepancy to various factors, including network depth and width, dataset complexity (Ainsworth et al., 2022) and high learning rates (Altintas et al.,

---

[*]Corresponding author: `ankit.sonthalia@uni-tuebingen.de`

2023). Theoretical investigation (Entezari et al., 2021; Ferbach et al., 2024) suggests that in general, the conjecture needs wide networks to hold.

In response to these findings, our research introduces the **star domain conjecture**. We propose that more generally, solutions in deep neural networks (DNNs) form a star domain rather than a convex set, modulo permutation symmetries. A *star domain* is a set $A$ with at least one special element, known as a *star point*, $a_0 \in A$ that is connected to every other element in $A$. A convex set is a specific instance of a star domain. The star domain conjecture thus proposes that in cases where convexity (Entezari et al., 2021) does not hold, a weaker form of convexity (*i.e.,* star-shaped connectivity) still exists.

The star domain conjecture is still a stronger assertion than mode connectivity (Garipov et al., 2018) which states that any two models $\theta_A$ and $\theta_B$ can be connected through a possibly non-linear path in the solution space. As a special case, this path could be as simple as a piece-wise linear path comprising a third point $\theta_C$ such that $(\theta_A, \theta_C)$ and $(\theta_B, \theta_C)$ are linearly connected. In contrast, our

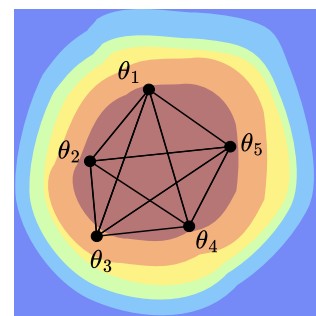
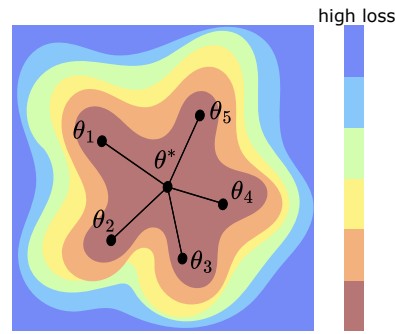

Convexity conjecture (Entezari et al.) holds only for wide networks.

Star domain conjecture (ours) holds even for narrower networks.

high loss

low loss

conjecture implies that *all* pairs of solutions are interconnected via a *shared* third solution, the star point, which is *common* to all solution pairs: $\exists \theta_C$ such that $\forall \theta_A, \theta_B \in S$, $(\theta_A, \theta_C)$ and $(\theta_B, \theta_C)$ are linearly connected, where $S$ is the solution set.

We substantiate our star domain conjecture with empirical evidence by introducing the Starlight algorithm to identify a candidate star model for a given learning task. Starlight finds a model that is linearly connected with a finite set of independent solutions. We demonstrate that these star model candidates have low loss barriers with an arbitrary set of solutions that were not used in constructing the star model candidates. This provides strong evidence that there exist star models that are linearly connected with other solutions.

In addition to validating the conjecture, our research delves into the distinctive characteristics of star models. We find that sampling from the star domain for Bayesian Model Averaging (BMA) leads to better uncertainty estimates than ensembles. Additionally, we demonstrate star models as a possible substitute to model ensembles, with lower inference time and memory footprint. These differences highlight the potential advantages of star models in various neural network applications.

We summarise our contributions:

1. The **star domain conjecture** for characterizing connectivity in neural network solution sets.

2. The **Starlight** algorithm for identifying a star model for a gradient-based learning task.

3. Analysis of practical benefits shown by the star models.

## 2 RELATED WORK

We introduce the relevant development of findings toward the understanding of DNN solution sets.

**Mode Connectivity.** Garipov et al. (2018) and Draxler et al. (2018) concurrently discovered mode connectivity. Gotmare et al. (2018) soon followed, showing non-linear connectivity even between networks obtained using different training schemes. Kuditipudi et al. (2019) explained mode connectivity via dropout stability and noise stability. Benton et al. (2021) went on to show that there exist not only simple paths, but also *volumes* of low loss, connecting several DNN solutions. These works focus on general, *non-linear* connectivity, while we study a stricter condition, *viz.,* linear connectivity.

**Linear Mode Connectivity (LMC).** Frankle et al. (2020) were the first to study LMC. Later, Entezari et al. (2021) proposed that SGD solution sets are convex modulo permutations, while Singh & Jaggi (2020); Ainsworth et al. (2022); Guerrero Peña et al. (2023) introduced "re-basin" methods, *i.e.,* methods for bringing different solutions into the same basin. Recent work (Ainsworth et al., 2022; Altintas et al., 2023; Benzing et al., 2022) also noted failure cases for LMC, while Ferbach et al. (2024) theoretically investigated convexity for sufficiently wide nets. Our analysis builds upon these findings and reveals evidence for a weaker property, *viz.,* star-shaped connectivity, in cases where convexity does not hold.

**Star-shaped connectivity (SSC).** Zhou et al. (2019) show that SGD follows an *epochwise star-convex path* but they focus on the optimization trajectory rather than the structure of the loss landscape. Annesi et al. (2023) provide valuable insights for SSC in the loss landscape for the simple case of the negative spherical perceptron. In contrast, we consider more complex models and learning tasks, and propose a novel verification method for SSC. Lin et al. (2024) explore star-shaped connectivity to *finitely many* solutions. Our work, in contrast, additionally considers permutation invariances (Ainsworth et al., 2022; Entezari et al., 2021) and provides evidence that star models trained this way might be connected to *infinitely many* other solutions.

**Practical Applications.** Mode connectivity has found applications in model fusion (Garipov et al., 2018; Singh & Jaggi, 2020), adversarial robustness (Zhao et al., 2019; Wang et al., 2023), continual learning (Mirzadeh et al., 2020; Wen et al., 2023), and federated learning (Wang et al., 2019; Ainsworth et al., 2022). In contrast, our work focuses on understanding the surface of the loss landscape. However, we also explore potential applications, *e.g.,* Bayesian Model Averaging.

## 3 THE STAR DOMAIN CONJECTURE

In this section, we introduce our star domain conjecture before discussing its practical applications in Bayesian model averaging and model ensembling in Section 4.

### 3.1 BACKGROUND: THE CONVEXITY CONJECTURE

Here, we formally state the convexity conjecture, starting with basic notations. A neural network is a function $f_\theta(\cdot)$ parameterized by $\theta \in \Theta$, where $\Theta$ is the parameter space. Given a dataset $\mathcal{D}$, we formulate a non-negative loss $\mathcal{L}(\theta) = \mathcal{L}(\theta; \mathcal{D}) \geq 0$ and minimize $\mathcal{L}(\theta)$ to find a solution in $\Theta$. The **solution set** is $S := \{\theta \mid \mathcal{L}(\theta) \approx 0\}$.

The **loss barrier** was first defined by Frankle et al. (2020). We use the formulation in Entezari et al. (2021), *i.e.,* the barrier between $\theta_A, \theta_B \in \Theta$ is $B(\theta_A, \theta_B) := \max_{t \in [0,1]} \widetilde{\mathcal{L}}_t(\theta_A, \theta_B)$, where

$$\widetilde{\mathcal{L}}_t(\theta_A, \theta_B) := \mathcal{L}((1-t) \cdot \theta_A + t \cdot \theta_B) - ((1-t) \cdot \mathcal{L}(\theta_A) + t \cdot \mathcal{L}(\theta_B)) \tag{1}$$

is the difference between the loss value at $t$, and the linear interpolation of the losses at the end-points. Two solutions $\theta_A, \theta_B \in \Theta$ are said to be **linearly mode-connected**, or **LMC** (Frankle et al., 2020), when their loss barrier is approximately zero: $B(\theta_B, \theta_A) \approx 0$.

The convexity conjecture is constructed upon a parameter space where the permutation symmetries are factored out. A **permutation invariance** (Brea et al., 2019) can be formulated as an equivalence relation $\sim$ between two points $\theta_A, \theta_B$ in the parameter space such that $\theta_A \sim \theta_B$ if and only if there exists a permutation $\pi$ of the parameters such that $\pi(\theta_A) = \theta_B$ *and* the functions represented by them are identical: $f_{\theta_A}(x) = f_{\theta_B}(x)$ for all $x$. Given two points $\theta_A$ and $\theta_B$, we look for the permutation of $\theta_B$ that connects it to $\theta_A$ (or vice versa) with as low a loss barrier as possible (Ainsworth et al., 2022; Entezari et al., 2021; Guerrero Peña et al., 2023). A **winning permutation** (Entezari et al., 2021) for models $\theta_A$ and $\theta_B$ is defined as

$$\pi_{\theta_A \to \theta_B} := \arg\min_{\pi \in \mathcal{P}_{\theta_A}} B(\pi(\theta_A), \theta_B) \tag{2}$$

where $\mathcal{P}_\theta := \{\pi \mid \pi(\theta) \sim \theta\}$ is the set of all function-preserving permutations of $\theta$.

**Conjecture 1.** *Convexity Conjecture (Entezari et al., 2021). Let $S$ be the set of SGD-reachable solutions for a deep neural network $f(\theta)$ trained for a certain task. Let $\theta_A, \theta_B \in S$ be two solutions. Then, there exists a minimum width $h$ such that if $f(\theta)$ is wider than $h$, then with high probability, $\theta_B$ can be permuted to obtain $\tilde{\theta}_B = \underset{\theta_B \to \theta_A}{\pi}(\theta_B)$ such that $\theta_A$ and $\tilde{\theta}_B$ are highly likely to be linearly mode-connected, i.e., $B\left(\tilde{\theta}_B, \theta_A\right) \approx 0$.*

We refer to this as the (quasi-) convexity conjecture, because, by definition, a convex set is precisely a set where the line segment between any two elements is included in the set. The conjecture provides a geometric intuition that the solution set is generally convex, modulo permutations.

Theoretical results only validate the conjecture in limited settings, given sufficiently wide networks (Entezari et al., 2021; Ferbach et al., 2024). Empirical validations exhibit mixed accounts. Ainsworth et al. (2022) notably achieve zero barrier between two ResNet-20-32 models trained on CIFAR-10, but there remains a loss barrier between narrower models, even after weight matching. They further report network depth and dataset complexity as aggravating factors. Benzing et al. (2022) provide interesting insights using their activation-matching permutation algorithm. While fully connected networks (FCNs) live in the same loss valley even at initialization, convolutional nets (CNNs) are usually not connected even after considering permutation invariances. Guerrero Peña et al. (2023) introduce Sinkhorn re-basin, a differentiable permutation-finding approach; however, even with two-layer NNs, the barrier between CIFAR-10 models, albeit low, remains non-zero. For CNN architectures like VGG, the barrier is substantially high. Altintas et al. (2023) show that aggravating factors for LMC include the Adam optimizer (Kingma & Ba, 2017), absence of warmup, and task complexity.

Hence, in cases where strong evidence for the convexity conjecture is absent, it is important to consider other possible topologies for general DNN solution sets. To this end, we propose the star domain conjecture for characterizing DNN solution sets that do not enjoy convexity (Entezari et al., 2021) modulo permutations.

### 3.2 THE STAR DOMAIN CONJECTURE

We propose a weaker form of convexity for characterizing DNN solution sets. We argue that DNN solution sets are generally *star domains*, modulo function-preserving permutations. While Annesi et al. (2023) demonstrate this property for simple spherical negative perceptrons (without permutations), we argue that it holds for even deeper, more complex nets after considering permutation invariances.

We start with the necessary definitions to make a formal description of the conjecture. A set $A \subset \mathbb{R}^n$ is a **star domain** if there exists an element $a_0 \in A$ such that for any other element $a \in A$ and $\forall t$ such that $0 \leq t \leq 1$, we have that $(1 - t) \cdot a_0 + t \cdot a \in A$, *i.e.,* all points on the line segment between $a_0$ and $a$ lie in $A$. We call such $a_0$ a **star point**. In the context of the parameter space, we refer to the star point of a star-domain-shaped solution set as a **star model**.

A "solution" in our work refers to global minima theoretically reachable by SGD, unless stated otherwise (please also refer to Appendix B.4).

**Conjecture 2.** *Star Domain Conjecture. Consider a neural network $f : \mathcal{X} \to \mathcal{Y}$, where $\mathcal{X} \subset \mathbb{R}^d$, $\mathcal{Y} \subset \mathbb{R}$. Assume that $f$ is parameterized by $\theta \in \mathbb{R}^m$. Define $S \subset \mathbb{R}^m$, the solution set of $f$ as the set of parameters such that every $\theta \in S$ minimizes a given loss $\mathcal{L}$ over a given dataset $\mathcal{D} = \{(x_i, y_i), x_i \in \mathcal{X}, y_i \in \mathcal{Y}, 1 \leq i \leq N\}$. Define $B(\theta_1, \theta_2)$ as the loss barrier encountered upon linearly interpolating between $\theta_1$ and $\theta_2$.*

*Then, for any $\epsilon > 0$, there exists a network width $h \in \mathbb{N}$ such that if $f(\cdot)$ is wider than $h$, then $S$ is a star domain modulo permutation symmetries, up to a tolerance of $\epsilon$. Specifically, there exists a star model $\theta^* \in S$ such that for any other solution $\theta \in S$, there exists a function-preserving permutation $\tilde{\theta} = \pi_{\theta \to \theta^*}(\theta)$ such that $B(\tilde{\theta}, \theta^*) < \epsilon$.*

A convex set is a special case of a star domain, where all the elements are star points. Thus, according to the convexity conjecture (Entezari et al., 2021), every member of a DNN solution set is a star model, given sufficient width. In contrast, the central thesis of our work is that as network width increases, star-domainness (where *some* models are star models) arises *before* convexity (where *all* models are star models, in agreement with the convexity conjecture).

Our conjecture concerns deep neural networks, where theoretical validation remains difficult. However, for the simpler case of two-layer neural networks with one-dimensional inputs, we state Theorem 1 here. We defer the proof of this theorem to Appendix A. Additionally, in Appendix A.2, we empirically validate our theoretical result under simple settings.

**Assumption 1.** *Consider a two-layer linear network $f_\theta(x) = \theta_1^\top \theta_2 x$, parameterized by $\theta = (\theta_1, \theta_2) \subset \Theta$ where $\theta_1, \theta_2 \in \mathbb{R}^m$. $m$ is the network width. Assume that the inputs are bounded $|x| \leq 1$. Define $\mathcal{L}(\theta; (x_i, y_i)_{i=1}^I) := \frac{1}{2|I|} \sum_i (\theta_1^\top \theta_2 x_i - y_i)^2$ to be the MSE loss. Let $S$ be the solution set with unit norms $\|\theta_1\|_2 = \|\theta_2\|_2 = 1$ that satisfy $\mathcal{L}(\theta) = 0$. Given two solutions $\phi, \theta \in S$, we define the barrier as $B(\phi, \theta) := \max_{t \in [0,1]} \mathcal{L}((1-t)\phi + t\theta) - (1-t)\mathcal{L}(\phi) - t\mathcal{L}(\theta)$.*

*Let $\lambda$ be the uniform distribution over the solution set $S$, which is well-defined as the solution set $S$ is measurable. We define the probability*

$$P(m, \epsilon) := \mathbb{P}_{\phi_1, \phi_2 \sim \lambda} \left( \forall \theta \in S, \exists \text{ permutation of neurons } \pi \text{ s.t. } B(\phi, \pi(\theta)) \leq \epsilon \right).$$

**Theorem 1.** *Given Assumption 1 and an arbitrary $\epsilon > 0$, $P(m, \epsilon) \to 1$ as $m \to \infty$.*

Theorem 1 guarantees the existence of star models for two-layer linear networks, under the constraints laid out in Assumption 1. This result implies that for a fixed tolerance level $\epsilon$, as $m$ increases, more and more models in the solution set become star models. Eventually, virtually all solutions will be star models, implying the convexity of the solution set (in agreement with the convexity conjecture).

### 3.3 FINDING A STAR MODEL

We provide empirical evidence for the star domain conjecture via two steps. First, we present a method for finding a star model. Second, we verify that the model found is indeed a star model: it has a low loss barrier with an arbitrary solution in $S$. Here, we focus on the first step.

We consider a necessary condition for a star model $\theta^\star$: given an arbitrary set of models $Z = \{\theta_1, \theta_2, \ldots, \theta_N\} \subset S$, $\theta^\star$ has to be connected to all of them, modulo permutation invariances.

We present a recipe for finding such a $\theta^\star$.

We first obtain a finite set $Z = \{\theta_1, \theta_2, \ldots, \theta_N\}$ of models, independently trained with different random seeds controlling the initialization, batch composition, and augmentation. We then formulate a loss function that, for fixed $Z$, encourages low loss barriers between $\theta$ and some permuted versions of $\{\theta_1, \theta_2, \ldots, \theta_N\}$. The objective may be expressed as

$$\theta_Z^\star = \arg\min_\theta \frac{1}{N} \sum_{\theta_n \in Z} B \left( \theta, \underset{\theta_n \to \theta}{\pi}(\theta_n) \right) \tag{3}$$

where $\underset{\theta_n \to \theta}{\pi}$ is the winning permutation defined in Section 3.1 that permutes $\theta_n$ without changing the represented function, and while minimizing the loss barrier against $\theta$. To solve this optimization problem, we propose to minimize the expected loss on the linear interpolation between the model in question $\theta$ and each source model $\theta_n$, after permutations. We modify the training objective as $\theta_Z^\star = \arg\min_\theta \widetilde{\mathcal{L}}_Z(\theta)$ where

$$\widetilde{\mathcal{L}}_Z(\theta) := \frac{1}{N} \sum_{n=1}^N \int_0^1 \mathcal{L} \left( (1-t) \cdot \theta + t \cdot \underset{\theta_n \to \theta}{\pi}(\theta_n) \right) dt \tag{4}$$

This expresses the expected loss on the set of line segments between $\theta$ and $\underset{\theta_n \to \theta}{\pi}(\theta_n)$, where each source model $\theta_n \sim \text{Unif}(Z)$ is chosen at random and then each point on the line segment is sampled as $t \in \text{Unif}[0, 1]$. The optimization problem in eq. (4) involves computational challenges. Resolving the continuous integral over $t$ is non-trivial for complex learning problems. Furthermore, $\underset{\theta_n \to \theta}{\pi}$ assumes access to the winning permutation. However, the winning permutation depends on $\theta$, which is constantly changing over the course of optimization. We introduce the following solutions.

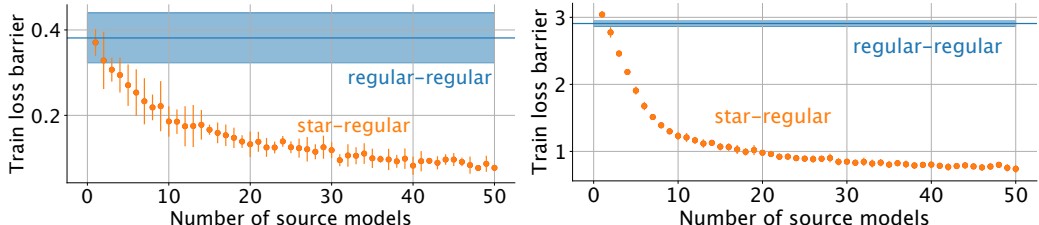

Figure 1: **Starness of a star model vs source models.** We plot the loss barriers $B(\theta^\star, \theta_h)$ between star models $\theta^\star$ and heldout models $\theta_h \in H$ at different numbers of source models $Z$ used for learning the star model $\theta^\star$ (orange points). The heldout set is disjoint with the source models: $H \cap Z = \emptyset$. We provide a reference point given by the loss barrier between two regular solutions $B(\theta_A, \theta_B)$ for $\theta_A, \theta_B \in S$ (blue plot). The error bars indicate one standard deviation across five held-out models $|H| = 5$. Incorporating more source models $|Z|$ enables finding a better star model with a lower loss barrier against an arbitrary solution.

**Monte-Carlo optimization scheme.** Instead of estimating $\widetilde{\mathcal{L}}_Z(\theta)$ precisely at every iteration, we rely on a Monte-Carlo estimation scheme, inspired by the parameter-curve fitting method by Garipov et al. (2018). At iteration $k \geq 1$, we sample $\theta_{n^{(k)}}$ uniformly from $Z$ and $t^{(k)}$ from Unif$[0, 1]$. Hence, we obtain a single point on the manifold, calculate the cross-entropy loss at this point, and subsequently the gradients for updating $\theta$.

---

**Algorithm 1** Starlight: Training a Star Model.

**Input.** dataset $\mathcal{D} = \{(x_i, y_i)\}_{i=1}^I$, source models $Z = \{\theta_1, \theta_2, \ldots, \theta_N\}$, initial model $\theta_0$, learning rate $\lambda$, number of batches $m$, number of steps $K$. Set $\theta \leftarrow \theta_0$.

**Output.** $\theta$

**for** $k = 1$ to $K$ **do**
  **if** $(k - 1) \mod m == 0$
    **for** $n = 1$ to $N$ **do**
      **Step 1.** Update $\theta_n \leftarrow \pi_{\theta_n \to \theta}(\theta_n)$
    **endfor**
  **endif**
  **Step 2.** Sample $\theta_n \sim \text{Unif}(Z)$, $t \sim \text{Unif}[0, 1]$, and a batch $\mathcal{B}$ from $\mathcal{D}$.
  **Step 3.** Compute loss $\mathcal{L}((1 - t) \cdot \theta + t \cdot \theta_n \,; \mathcal{B})$.
  **Step 4.** Compute gradients $v \leftarrow \nabla_\theta \mathcal{L}((1 - t) \cdot \theta + t \cdot \theta_n)$.
  **Step 5.** Update $\theta \leftarrow \theta - \lambda(1 - t) \cdot v$.
**endfor**

---

**Finding optimal permutations.** We perform weight matching (Ainsworth et al., 2022), *i.e.,* we seek a permutation $\pi_n$ that maximizes the dot product $\theta \cdot \pi_n(\theta_n)$, for each $\theta_n \in Z$. This procedure aligns each source model $\theta_n$ with the candidate star model $\theta$. This operation is performed at the beginning of every *epoch* instead of every *iteration*, speeding up the optimization process significantly.

Algorithm 1 describes the detailed procedure. Once we find a $\theta$ that has a low expected loss $\widetilde{\mathcal{L}}_Z(\theta)$ on the linear paths to a finite set of source models $Z$, we may verify if this $\theta$ is likewise linearly connected with an arbitrary solution $\theta_{N+1} \notin Z$.

### 3.4 EMPIRICAL EVIDENCE

We introduced Starlight to find a candidate star model. Now, we propose a method to verify if the model found in Section 3.3 is a star model by checking its linear connection to an arbitrary solution $\theta_{N+1} \notin Z$, *i.e.,* not part of the set of source models used for finding the star model. We refer to such models as *held-out* solutions $H$ that are disjoint from the source models: $H \cap Z = \emptyset$.

We describe our main findings with reference to ResNet-18 (He et al., 2016) models trained on CIFAR (Krizhevsky et al., 2012) using SGD, using 50 source models and 5 held-out models. We present results for additional architectures (*e.g.,* VGG (Simonyan & Zisserman, 2015) and DenseNet (Huang et al., 2017)), a large-scale dataset (ImageNet-1k (Deng et al., 2009)) and settings (for instance, Adam

Table 1: **Empirically verifying the star domain conjecture.** "Regular loss" and "Star loss" indicate training losses for regular models in $Z$ and star models $\theta^\star$, respectively. "Star-regular" refers to the barrier $B(\theta^\star, \theta_h)$ between a star model and one of the heldout models in $H$. For comparison, "Regular-regular" is the loss barrier $B(\theta_A, \theta_B)$ between two arbitrary models. We report values up to one standard deviation over several runs, except for ImageNet. In each case, star models exhibit significantly lower loss barriers with other models, than the corresponding average loss barrier between two regular models.

| Dataset | Architecture | Regular loss | Star loss | Regular-regular | Star-regular |
|---------|--------------|--------------|-----------|-----------------|--------------|
| CIFAR-10 | ResNet-18 | $0.001 \pm 0.000$ | $0.001 \pm 0.000$ | $0.383 \pm 0.056$ | $0.078 \pm 0.007$ |
| CIFAR-10 | ResNet-18 (Adam) | $0.001 \pm 0.000$ | $0.015 \pm 0.000$ | $1.368 \pm 0.551$ | $0.335 \pm 0.022$ |
| CIFAR-10 | VGG11 | $0.003 \pm 0.000$ | $0.022 \pm 0.000$ | $0.515 \pm 0.034$ | $0.131 \pm 0.005$ |
| CIFAR-10 | VGG19 | $0.001 \pm 0.000$ | $0.059 \pm 0.000$ | $1.281 \pm 0.153$ | $0.336 \pm 0.078$ |
| CIFAR-10 | DenseNet | $0.001 \pm 0.000$ | $0.157 \pm 0.000$ | $4.634 \pm 0.727$ | $1.729 \pm 0.409$ |
| CIFAR-100 | ResNet-18 | $0.004 \pm 0.001$ | $0.005 \pm 0.000$ | $2.905 \pm 0.047$ | $0.756 \pm 0.049$ |
| CIFAR-100 | DenseNet | $0.006 \pm 0.000$ | $0.635 \pm 0.000$ | $6.920 \pm 0.216$ | $3.735 \pm 0.180$ |
| ImageNet-1k | ResNet-18 | $0.711$ | $1.380$ | $5.948$ | $2.794$ |

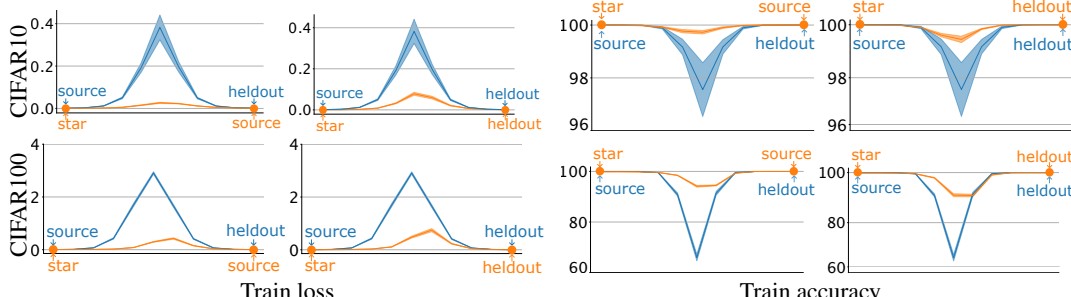

Figure 2: **Loss barriers for star models**. We interpolate between a star model $\theta^\star$ and regular models that are trained with SGD. There are two types of regular models, depending on whether they are used for finding the star model: source models $Z$ are used, and heldout models $H$ are not. Along the interpolation, we visualize the loss barrier by plotting the loss and accuracy values (orange curves). For these curves, $t = 0$ corresponds to the star model $\theta^\star$. For reference, we plot the interpolation between two arbitrary regular models (blue curves). The error bands correspond to one standard deviation.

(Kingma & Ba, 2017)) in Table 1 and Appendix E. Likewise, our empirical findings are built upon the training loss and accuracy, but we confirm that they also transfer over to test loss and accuracy in Appendix D.1. We largely use standard recipes to train the models in our experiments, with the exception of star models where we additionally incorporate the steps in Algorithm 1. We further describe our experimental setup in Appendix B. We summarize our observations below.

**Convexity conjecture does not hold.** In Figure 2, we show loss barriers between two independently trained solutions (blue "regular-regular" curves). We observe that the loss increases and accuracy drops significantly at around $t = 0.5$, even after applying the algorithm (Ainsworth et al., 2022) to find the winning permutation. We present another piece of evidence that the convexity conjecture does not hold for thin ResNets, reconfirming the findings of Ainsworth et al. (2022).

**Star model has low loss barriers with other solutions.** In Figure 2, we show the training losses and accuracies along linear paths between the candidate star model $\theta^\star$ and other types of solutions (either source models $Z$ or held-out models $H$). They are indicated with red curves. As a reference, we always plot the confidence interval of loss and accuracy values along the line segments between two regular solutions (blue curves). We observe that, for the source models in $Z$, star-to-regular connections enjoy essentially zero loss barriers, in contrast with regular-to-regular connections, which remain significantly higher at $0.381$, for CIFAR-10. This demonstrates that it is possible to find a model $\theta^\star$ connected to $|Z| = 50$ models simultaneously. The same is true for the line segments between the star model $\theta^\star$ and a held-out model picked from $|H| = 5$ models; although the barrier

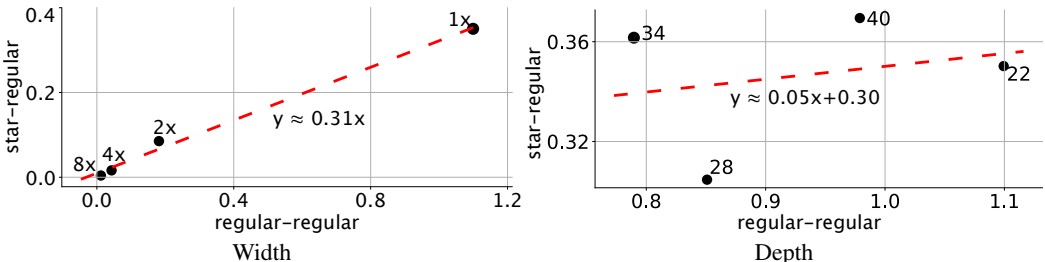

Figure 3: **"Starness" vs. model width and depth**. For starness vs. model width (left), we vary the width of a WideResNet (depth 22) from $1\times$ to $8\times$. For starness vs. model depth, we vary the depth of a WideResNet (width $1\times$) from 22 to 40 layers. For each depth-width combination, we plot the loss barriers $B(\theta^\star, \theta_h)$ between star models $\theta^\star$ and heldout models $\theta_h \in H$ on the y-axis. As a reference point, we plot the barrier between two regular solutions $B(\theta_A, \theta_B)$, on the x-axis. The points are annotated with the corresponding widths or depths. Star models consistently enjoy better linear connections with regular models, than do the regular models amongst each other.

between the star model and the heldout model is non-zero, it remains as low as $0.077$ compared to $0.381$ for the regular-to-regular case.

**A greater number of source models enhances "starness".** Our star model is constructed from the set of source models $Z$. We question whether greater $|Z|$ induces greater "starness" of the solution found by Starlight. In Figure 1, we plot the loss barrier against the number of source models $2 \le |Z| \le 50$ used to construct the star model. For statistical significance, we include loss barrier statistics between two regular, independently trained models in $S$ with error bars indicating one standard deviation. We observe that the loss barriers between these star models and the held-out models decrease as $|Z|$ increases. The decreasing trend has not saturated after $|Z| = 50$. We stopped there because of computational limits. However, including more source models is likely to enhance connectivity between the obtained star model and the other solutions even further.

**Effect of model width and depth.** Prior work stresses the importance of model width and depth (Ainsworth et al., 2022; Entezari et al., 2021) in determining loss barriers between two solutions. We investigate the effect of model width and depth for residual nets. Specifically, we consider WideResNets (Zagoruyko & Komodakis, 2017) of widths $1\times$, $2\times$, $4\times$, and $8\times$ that of a normal ResNet (depth 22). We also consider ResNets of depths 22, 28, 34, and 40. We compare the barriers achieved by "regular-regular" and "star-regular" pairs for each case Figure 3. Our investigation confirms existing reports of decreasing loss barriers as model width increases. We observe significantly lower star-regular barriers than regular-regular barriers for models of identical widths (*e.g.,* roughly $0.004$ compared to $0.012$ at width $8x$). In fact, it is possible to fit a linear regression line to the observed barrier values, wherein the star-regular barriers are about a third of the regular-regular barriers at any given width (Figure 3, left). We draw similar conclusions from varying depth (Figure 3, right), although the change in barriers as we change the model depth is not quite as pronounced as it is for the varying width case.

**Effect of optimizer.** While both the convexity conjecture and the star domain conjecture involve solution sets obtained through SGD, we also investigate the impact of using the Adam optimizer. Specifically, we train 15 regular models, with $|H| = 5$ and ($|Z| = 10$). We then train a star model and evaluate its barriers with the models in the held-out set. Results can be found in Table 1 (second row). We observe that Adam-trained regular solutions have a higher loss barrier between them ($1.368$) compared to SGD-trained regular solutions ($0.383$). Likewise, the barrier between the star model and regular models also increases from $0.078$ for SGD solutions to $0.335$ for Adam solutions. While both "regular-regular" and "star-regular" connections suffer with this change of optimizer, "star-regular" connections still fare significantly better than "regular-regular" connections. This finding suggests that Adam solutions are also highly likely to enjoy star-shaped connectivity.

**Caveats**. Despite the promising observations above, our star domain conjecture lacks theoretical verification, except for the simple case of two-layer linear networks (Appendix A). Hence, for deep neural networks, it remains a conjecture. From the empirical perspective, loss barriers between the star model and other solutions often yield values that are significantly greater than zero. However, we

Table 2: **Model fusion performances**. "Regular" indicates single models; Ensemble indicates a vanilla average of the probability vectors across the member models; "Star" indicates a model found using Starlight using the regular models as the source set $Z$. ResNet18 has been used throughout. We show one standard deviation for the error bars. In addition, we report the accuracy of the best member in the ensemble ("Best of $n$") and the accuracy of the best star model ("Best of 3"). Star models perform better than single, regular models but use only a fraction of the compute required by the ensemble at test time.

| Dataset | #Models | Regular | Best of $n$ | Ensemble | Star | Best of 3 |
|---|---|---|---|---|---|---|
| | 2 | $95.21 \pm 0.03$ | 95.24 | 95.76 | $95.30 \pm 0.16$ | 95.43 |
| CIFAR-10 | 5 | $95.07 \pm 0.14$ | 95.24 | 96.02 | $95.17 \pm 0.15$ | 95.27 |
| | 50 | $95.13 \pm 0.16$ | 95.44 | 96.27 | $95.32 \pm 0.20$ | 95.54 |
| | 2 | $77.32 \pm 0.16$ | 77.49 | 79.58 | $77.96 \pm 0.24$ | 78.14 |
| CIFAR-100 | 5 | $77.36 \pm 0.21$ | 77.68 | 80.38 | $78.12 \pm 0.02$ | 78.15 |
| | 50 | $77.33 \pm 0.28$ | 77.94 | 81.30 | $78.38 \pm 0.10$ | 78.48 |
| Train / test complexity | | $\mathcal{O}(1)/\mathcal{O}(1)$ | | $\mathcal{O}(n)/\mathcal{O}(n)$ | $\mathcal{O}(n)/\mathcal{O}(1)$ | |

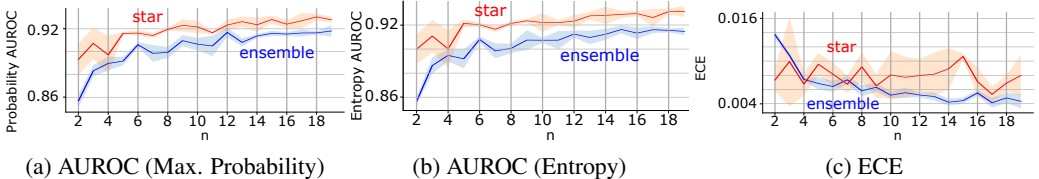

(a) AUROC (Max. Probability)     (b) AUROC (Entropy)     (c) ECE

Figure 4: **Bayesian model averaging**. The star model was trained using 50 source models. The x-axis denotes the number of models sampled from the star domain for Bayesian model averaging or from the set of source models.

emphasize that this paper is focused on providing the lower bound in evidence supporting the star domain conjecture. Considering a larger number of source models for the star model construction, improving Starlight, and developing a better algorithm for finding the winning permutations will potentially contribute to the discovery of better star models in the solution set.

**Conclusion**. Our experimental results confirm existing reports that the convexity conjecture requires very wide networks to hold, and has otherwise several failure cases for which we propose a relaxed version, *viz.,* the star domain conjecture. We obtain strong empirical evidence that the star model found through Starlight is likely to be a true star model. Our analysis thus sheds further light on solution set geometry for narrower and deeper networks, as well as for complex learning tasks where the convexity conjecture struggles. We invite the community to expand upon our findings and converge toward a more accurate understanding of the loss landscape.

## 4 PRACTICAL APPLICATIONS

The star domain conjecture introduces a novel dichotomy of solution types: "star" and "non-star" models. Most solutions are non-star and lack linear connections with other solutions. However, in Section 3, we have presented strong evidence for the existence of star models. In this section, we examine the properties and potential benefits of star models in practice. Section 4.1 explores whether star models and the surrounding star domain provide a better posterior for Bayesian Model Averaging. In Section 4.2, we propose star models as a practical alternative to model ensembling.

### 4.1 BAYESIAN MODEL AVERAGING

Bayesian model averaging (BMA) enhances uncertainty estimation by averaging predictions from the posterior of models in the parameter space. Posterior families in the literature range from simple Gaussian (Blundell et al., 2015) and Bernoulli (Gal & Ghahramani, 2016) distributions to more

complex geometries like splines (Garipov et al., 2018) and simplices (Benton et al., 2021). Here, we examine if the star domain provides a good posterior family for BMA-based uncertainty estimation.

**Setup**. The posterior of interest is the collection of line segments between the star model $\theta^\star$ and other solutions $\{\theta_1, \cdots, \theta_N\}$ that are independently found. Similarly to Starlight, we sample first from the model index of $\{1, \cdots, N\}$ uniformly and then sample from the line segment Unif$[0, 1]$. As in standard BMA, we consider a set of models sampled from the posterior and the post-softmax average of their outputs. We use ResNet-18 models trained on CIFAR-10. As a baseline, we present the BMA for the independent solutions $\{\theta_1, \cdots, \theta_N\}$.

**Evaluation**. We assess the predictive uncertainty of the BMA-based confidence estimates. For the ranking metric, we use the area under ROC curve (AUROC), considering both max-probability and entropy-based confidence measures. We also show results based on expected calibration error (ECE).

**Results**. Figure 4 shows uncertainty quantification at different numbers of posterior samples from 2 to 19. BMA using the star domain posterior consistently exhibits better AUROC values than baseline deep-ensemble estimates. However, ECE is worse than that of the deep ensemble. The star domain posterior provides avenues for more precisely ranked uncertainty estimates, albeit absolute-value uncertainty quantification may not be precise.

**Conclusion**. Our proposed star domain posterior offers better uncertainty estimates than the deep ensemble baseline in rank-based predictive uncertainty evaluation.

## 4.2 POTENTIAL USAGE IN MODEL FUSION

Given a fixed amount of training data, a popular approach to maximize model generalizability is ensembling, *i.e.,* fusing predictions from multiple independent models. This basic approach suffers from computational complexities during both training and inference. Every input has to be processed by individual member models at test time. Storing multiple models also leads to a higher memory footprint, scaling linearly with the number of ensemble members.

Starlight can also be understood as a method for aggregating multiple source models $Z = \{\theta_1, \cdots, \theta_N\}$ into a single model $\theta^\star$. From a computational perspective, star models reduce the necessary time and storage complexity during inference. We investigate whether the star models provide an enhanced generalization compared to the individual models.

**Setup**. We slightly modify the training objective of Starlight to align it with a better generalization capability of the star model. We add a cross-entropy term $\mathcal{L}(\theta)$ so that $\mathcal{L}_{\text{total}}(\theta, Z) = \widetilde{\mathcal{L}}_Z(\theta) + \mathcal{L}(\theta)$, where $\widetilde{\mathcal{L}}_Z(\theta)$ is the original optimization objective for the star model discovery in eq. (4).

**Evaluation**. We evaluate test accuracies for star models trained with varying numbers of source models ($|Z|$) and compare them to ensembles using the same source models.

**Results**. Results in Table 2 show that star models consistently outperform regular models ($78.4\%$ vs. $77.3\%$) for CIFAR-100 with $|Z| = 50$). While less accurate than ensembles over $Z$, star models require only a fraction of the compute during inference.

**Conclusion**. "Starness" of a solution may enhance generalization. In scenarios where test-time inference costs are critical, star models could be a promising alternative to vanilla ensembles.

## 5 CONCLUSION

This paper proposes a novel understanding of SGD loss landscapes. The traditional picture before Garipov et al. (2018) was one of extreme non-convexity, in contrast with the current picture of near-perfect convexity in a canonical, modulo-permutations space (Entezari et al., 2021) for extremely wide nets. Our claim becomes relevant when narrower and deeper nets, complex datasets, and different optimization schemes are considered. We propose a weaker form of convexity in these cases, *i.e.,* the solution set is a star domain modulo permutations. Our empirical findings support this hypothesis. We propose the Starlight algorithm to find candidate "star models" and verify that they are indeed linearly connected to other solutions. In addition to the empirical evidence for the star domain conjecture, we present potential use cases for star models in practice, including uncertainty estimation through Bayesian model averaging, and model fusion.

ACKNOWLEDGEMENTS

This work was supported by the German Federal Ministry of Education and Research (BMBF): Tübingen AI Center, FKZ: 01IS18039A. The authors thank the International Max Planck Research School for Intelligent Systems (IMPRS-IS) for supporting Alexander Rubinstein. The authors would also like to thank Arnas Uselis and Bálint Mucsányi for helpful insights.

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

# A  SPECIAL CASE STUDY: L-LAYER LINEAR NETWORKS

Here, we analyse the case of an $L$-layer linear network, focusing on $L = 2$. This relatively simpler case enables us to perform theoretical analysis that would have been significantly more challenging for deep neural networks.

Our theoretical and empirical analyses suggest that the geometry of the solution set for 2-layer linear networks is approximately star-shaped. More precisely, the solution set of such a network will have increasingly many star models as the width increases; eventually, at sufficient width, almost all models are star models, naturally implying convexity of the solution set. As promised in Section 3.2, we share our theoretical and empirical findings in detail below.

## A.1  THEORETICAL VERIFICATION

We re-state and prove our claim for 2-layer linear networks (Theorem 1).

**Assumption 2.** *(re-stating Assumption 1) Consider a two-layer linear network $f_\theta(x) = \theta_1^\top \theta_2 x$, parameterized by $\theta = (\theta_1, \theta_2) \subset \Theta$ where $\theta_1, \theta_2 \in \mathbb{R}^m$. $m$ is the network width. Assume that the inputs are bounded $|x| \leq 1$. Define $\mathcal{L}(\theta; (x_i, y_i)_{i=1}^I) := \frac{1}{2|I|} \sum_i (\theta_1^\top \theta_2 x_i - y_i)^2$ to be the MSE loss. Let $S$ be the solution set with unit norms $\|\theta_1\|_2 = \|\theta_2\|_2 = 1$ that satisfy $\mathcal{L}(\theta) = 0$. Given two solutions $\phi, \theta \in S$, we define the barrier as $B(\phi, \theta) := \max_{t \in [0,1]} \mathcal{L}((1-t)\phi + t\theta) - (1-t)\mathcal{L}(\phi) - t\mathcal{L}(\theta)$.*

*Let $\lambda$ be the uniform distribution over the solution set $S$, which is well-defined as the solution set $S$ is measurable. We define the probability*

$$P(m, \epsilon) := \mathbb{P}_{\phi_1, \phi_2 \sim \lambda} \left( \forall \theta \in S, \exists \text{ permutation of neurons } \pi \text{ s.t. } B(\phi, \pi(\theta)) \leq \epsilon \right) \tag{5}$$

**Theorem 2.** *(re-stating Theorem 1) Given Assumption 2 and an arbitrary $\epsilon > 0$, $P(m, \epsilon) \to 1$ as $m \to \infty$.*

*Proof.* Observe that the output of the dot product $\theta_1^\top \theta_2$ is a scalar $C$. Considering the MSE loss defined in terms of the dot product, $\sum_i (Cx_i - y_i)^2$, the solution is unique due to strong convexity. We may thus write down the solution set as $\{(\theta_1, \theta_2) | \theta_1 \theta_2 = C\}$ where $C$ minimises the MSE loss.

Now, define

$$Q(m, \epsilon) := \mathbb{P}_{\phi_1, \phi_2 \sim \lambda} \left( \forall \theta \in S, \exists \text{ permutation of neurons } \pi \text{ s.t. } |\phi_1 \pi(\theta_2) + \pi(\theta_1)\phi_2 - 2C|^2 \leq \epsilon \right) \tag{6}$$

and prove the lemma below:

**Lemma 1.** *Assume the conditions of Assumption 2. Then for any $\epsilon > 0, m \in \mathbb{N}$, we have that $P(m, \epsilon) \geq Q(m, \epsilon)$.*

It follows that it is sufficient to prove: $Q(m, \epsilon) \to 1$ as $m \to \infty$ for all $\epsilon > 0$.

**Lemma 2.** *Consider the solution set $S := \{(\phi_1, \phi_2) \,|\, \phi_1, \phi_2 \in \mathbb{R}^m, \phi_1^\top \phi_2 = C, \|\phi_1\|_2 = \|\phi_2\|_2 = 1\}$, where $|C| \leq 1$. Let $\lambda$ be the Lebesgue measure confined to $S \subset \mathbb{R}^m \times \mathbb{R}^m$. Let $\epsilon > 0$. Then,*

$$\mathbb{P}_{\phi_1, \phi_2 \sim \lambda} \left[ \sup_{\theta_1, \theta_2 : \theta_1^\top \theta_2 = C} \min_{\pi \in S_n} \left| \phi_1^\top \pi(\theta_2) + \pi(\theta_1)^\top \phi_2 - 2C \right|^2 \leq \epsilon \right] \to 1$$

*as $m \to \infty$, where $S_m$ is the set of permutations on $\{1, \cdots, m\}$.*

Proving the lemmas will conclude the proof. $\square$

Now, let us prove the lemmas.

**Lemma 1.** *Assume the conditions of Assumption 2. Then for any $\epsilon > 0$ and $m \in \mathbb{N}$, we have that $P(m, \epsilon) \geq Q(m, \epsilon)$.*

*Proof.* Fix the width $m$ and tolerance $\epsilon > 0$. We show that for all solutions $\theta \in S$, there exists a permutation $\pi$ such that

$$|\phi_1^\top \pi(\theta_2) + \pi(\theta_1)^\top \phi_2 - 2C|^2 < \epsilon, \tag{7}$$

then $\phi$ is a star model with tolerance $\epsilon$. Specifically, we will prove that if equation 7 holds, then

$$B(\phi, \theta) < \epsilon. \tag{8}$$

This implies that the event of $\phi$ satisfying equation 7 is a subset of the event of $\phi$ being a star model, proving that $P(m, \epsilon) \geq Q(m, \epsilon)$.

Let $x$ be an input and $y$ be the corresponding label. Since $\phi$ and $\pi(\theta)$ are solutions, $\mathcal{L}(\phi, x) = \mathcal{L}(\pi(\theta), x) = 0$. We are interested in the linear interpolation between $\phi = (\phi_1, \phi_2)$ and $\pi(\theta) = (\pi(\theta_1), \pi(\theta_2))$ in the space $\mathbb{R}^m \times \mathbb{R}^m$. For any $t \in [0, 1]$, consider:

$$((1-t)\phi_1 + t\pi(\theta_1))^\top ((1-t)\phi_2 + t\pi(\theta_2))x = (1-t)^2 \phi_1^\top \phi_2 x + t(1-t)(\phi_1^\top \pi(\theta_2)x + \pi(\theta_1)^\top \phi_2 x) + t^2 \pi(\theta_1)^\top \pi(\theta_2)x$$
$$= (1 - 2t + 2t^2)y + t(1-t)(\phi_1^\top \pi(\theta_2)x + \pi(\theta_1)^\top \phi_2 x)$$
$$:= f_t(\phi, \pi(\theta), x),$$

where we used $y = \phi_1^\top \phi_2 x = \pi(\theta_1)^\top \pi(\theta_2)x$.

Define the loss at interpolation $t$:

$$\mathcal{L}_t(\phi, \pi(\theta), x) := [f_t(\phi, \pi(\theta), x) - y]^2.$$

Then,

$$\mathcal{L}_t(\phi, \pi(\theta), x) = \left[ (-2t + 2t^2)y + t(1-t)(\phi_1^\top \pi(\theta_2)x + \pi(\theta_1)^\top \phi_2 x) \right]^2$$
$$= t^2(1-t)^2 \left[ \phi_1^\top \pi(\theta_2)x + \pi(\theta_1)^\top \phi_2 x - 2y \right]^2$$
$$= t^2(1-t)^2 \left[ \phi_1^\top \pi(\theta_2) + \pi(\theta_1)^\top \phi_2 - 2C \right]^2 \|x\|^2$$
$$\leq |\phi_1^\top \pi(\theta_2) + \pi(\theta_1)^\top \phi_2 - 2C|^2 \quad \text{(since } 0 \leq t \leq 1 \text{ and } \|x\| \leq 1\text{)}$$
$$< \epsilon \quad \text{(by assumption)}.$$

Thus, for all $t \in [0, 1]$,
$$\mathcal{L}_t(\phi, \theta') = \mathbb{E}_{x \in \mathcal{X}}[\mathcal{L}_t(\phi, \theta', x)] < \epsilon.$$

Now, consider the function $B(\phi, \theta')$:

$$B(\phi, \theta') := \max_{t \in [0,1]} \left[ \mathcal{L}_t(\phi, \theta') - (1-t)\mathcal{L}(\phi) - t\mathcal{L}(\theta') \right]$$
$$= \max_{t \in [0,1]} \mathcal{L}_t(\phi, \theta') \quad \text{(since } \mathcal{L}(\phi) = \mathcal{L}(\theta') = 0\text{)}$$
$$< \epsilon.$$

Therefore, $B(\phi, \theta') < \epsilon$, which implies that $\phi$ is a star model with tolerance $\epsilon$. $\qquad\square$

**Lemma 2.** *Consider the solution set $S := \{(\phi_1, \phi_2) \mid \phi_1, \phi_2 \in \mathbb{R}^m, \phi_1^\top \phi_2 = C, \|\phi_1\|_2 = \|\phi_2\|_2 = 1\}$, where $|C| \leq 1$. Let $\lambda$ be the uniform distribution confined to $S \subset \mathbb{R}^m \times \mathbb{R}^m$. Let $\epsilon > 0$. Then,*

$$\mathbb{P}_{\phi_1, \phi_2 \sim \lambda} \left[ \max_{(\theta_1, \theta_2) \in S} \min_{\pi \in S_n} \left| \phi_1^\top \pi(\theta_2) + \pi(\theta_1)^\top \phi_2 - 2C \right|^2 \leq \epsilon \right] \to 1$$

*as $m \to \infty$, where $S_m$ is the set of permutations on $\{1, \cdots, m\}$.*

*Proof.* We may sample $\phi_1, \phi_2 \sim p_\lambda(\phi_1, \phi_2)$ by sampling from $p_\lambda(\phi_1)p_\lambda(\phi_2|\phi_1)$. Note that $p_\lambda(\phi_1) = \text{Unif}\{\|\phi_1\|_2 = 1\}$ and $p_\lambda(\phi_2|\phi_1) = \text{Unif}\{\|\phi_2\|_2 = 1 \text{ and } \phi_1^\top \phi_2 = C\}$.

Then, each element $\phi_{1i}$ (for $i \in \{1, \cdots, m\}$) follows the distribution defined by the PDF: $p(\phi_{1i}) = C_n(1 - \phi_{1i}^2)^{\frac{n-3}{2}}$ for some constant $C_n$ and $\phi_i \in [-1, 1]$. $p(\phi_{1i})$ converges in distribution to $N(0, \frac{1}{m})$ as $n \to \infty$. As such, given an arbitrary $\delta > 0$, we may choose $n$ such that the CDF of $\phi_{1i}$ is uniformly bounded away from the CDF of $N(0, \frac{1}{m})$ by $\frac{\epsilon}{3}$. Note that given samples $\phi_{1i}$ from $p_\lambda(\phi_1)$, we may sort them $(\phi_{1\sigma(1)}, \cdots, \phi_{1\sigma(m)})$ with some suitable permutation $\sigma \in S_m$. This lets us build

the empirical CDF defined by $F_{\phi_1(x)} := \frac{1}{n} \sum_i 1_{x \geq \phi_{1\sigma(i)}}$. Given two samples $\phi_1 \sim p_\lambda(\phi_1)$ and $\theta_1 \sim p_\lambda(\theta_1)$, we may sort them with permutations $\sigma$ and $\tau$, respectively, to ensure:

$$\|\sigma(\phi_1) - \tau(\theta_1)\|_2 \leq \|\sigma(\phi_1) - F_{N(0,1/n)}\|_2 + \|\tau(\theta_1) - F_{N(0,1/n)}\|_2 \leq \frac{\epsilon}{3} + \frac{\epsilon}{3} \qquad (9)$$

with high chance $1 - \delta$.

Taking $\pi = \sigma^{-1} \circ \tau$, we have

$$
\begin{aligned}
\left|\phi_1^\top \pi(\theta_2) + \pi(\theta_1)^\top \phi_2 - 2C\right|^2 &\leq \left|\phi_1^\top \pi(\theta_2) - C\right|^2 + \left|\pi(\theta_1)^\top \phi_2 - C\right|^2 \\
&\leq \left|\phi_1^\top \pi(\theta_2) - \pi(\theta_1)^\top \pi(\theta_2)\right|^2 + \left|\pi(\theta_1)^\top \pi(\theta_2) - C\right|^2 \\
&\quad + \left|\pi(\theta_1)^\top \phi_2 - \phi_1^\top \phi_2\right|^2 + \left|\phi_1^\top \phi_2 - C\right|^2 \\
&\leq \left|\phi_1^\top \pi(\theta_2) - \pi(\theta_1)^\top \pi(\theta_2)\right|^2 + \left|\pi(\theta_1)^\top \phi_2 - \phi_1^\top \phi_2\right|^2 \\
&\leq \|\phi_1 - \pi(\theta_1)\|_2^2 \|\pi(\theta_2)\|_2^2 + \|\pi(\theta_1) - \phi_1\|_2^2 \|\phi_2\|_2^2 \\
&< \frac{\epsilon}{3} + \frac{\epsilon}{3} < \epsilon
\end{aligned}
$$

with high chance $1 - \delta$.

This proves that

$$\mathbb{P}_{\phi,\theta\sim\lambda} \left[\min_{\pi \in S_m} \left|\phi_1^\top \pi(\theta_2) + \pi(\theta_1)^\top \phi_2 - 2C\right|^2 \leq \epsilon\right] \to 1$$

as $m \to \infty$. This result can easily be extended to the statement in the lemma using the compactness of the unit sphere. $\qquad\square$

## A.2 EMPIRICAL VERIFICATION

In the theoretical analysis above, we hypothesized for two-layer linear networks that as network width gets larger, the proportion of star models in the solution set increases. To verify this, we performed a small-scale analysis using 2-layer linear networks. First, we provide empirical evidence towards Lemma 2 in Appendix A.2.1. Then, in Appendix A.2.2, we consider a full-fledged learning problem and show that the proportion of star models increases as network width increases.

### A.2.1 DATA-FREE VERIFICATION

We provide empirical evidence towards Lemma 2 by considering vectors confined to the surface of a unit sphere.

**Setup.** We consider samples of $\phi = (\phi_1, \phi_2)$ and $\theta = (\theta_1, \theta_2)$ where $\phi_1, \phi_2, \theta_1, \theta_2 \in \mathbb{R}^n$, and $\|\phi_1\|_2 = \|\phi_2\|_2 = \|\theta_1\|_2 = \|\theta_2\|_2 = 1$, $\phi_1^\top \phi_2 = \theta_1^\top \theta_2 = C$. Specifically, we consider $C = 0.5$. The sampling is carefully carried out to reflect the conditions assumed in Lemma 2. We consider different values of $\epsilon$ and compute $Q(m, \epsilon)$, *i.e.,* the number of samples $\phi$ out of $N_\phi$ for which the condition in Lemma 2 holds, as $m$ increases. We set $N_\phi = N_\theta = 100$. The width $m$ ranges from $2^1$ to $2^{10}$, while we consider 50 values of $\epsilon$ between 0.01 and 0.1 (inclusive). Note that we use the same $\theta$ samples for evaluating all $\phi$ samples.

**Results and conclusion.** We present the results in Figure 5. We observe that as width (marked along the x-axis) increases, $Q(m, \epsilon)$ increases as well. Before we reach $Q(m, \epsilon) = 1$, we (almost) always pass through an intermediate region where $0 < Q(m, \epsilon) < 1$, thus confirming that $Q(m, \epsilon)$ slowly grows to 1 as width increases. Thus, we successfully validate Lemma 2.

### A.2.2 VERIFICATION OF MAIN THEOREM

In the previous section, we validated Lemma 2 for the case where the input dimensionality $d = 1$. Here, we further validate the statement of Theorem 2 for the case where $d = 2$, using a full-fledged machine learning task. Note that our experimental setup slightly differs from the conditions of Theorem 1 (*e.g.,* binary cross-entropy loss instead of MSE, and $d = 2$ instead of $d = 1$). This is on purpose: we aim to show that our theoretical result holds even after relaxing the conditions therein.

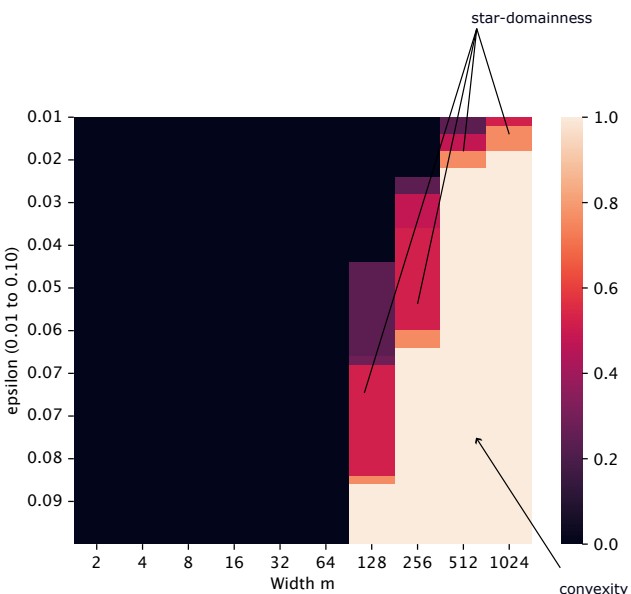

Figure 5: **Verification of our theoretical result in Lemma 2**. We compute $Q(m, \epsilon)$ at different widths $m$ and tolerances $\epsilon$. The y-axis denotes the tolerances, ranging from $0.01$ at the top to $0.1$ at the bottom. The x-axis denotes the widths $m$. The color bar encodes different values of $Q(m, \epsilon)$, ranging from $0$ to $1$. For fixed $\epsilon$, as width $m$ increases and $Q(m, \epsilon)$ goes from $0$ to $1$, there almost always exists a transitionary period where $0 < Q(m, \epsilon) < 1$, *i.e., some, but not all* samples $\phi \sim \lambda$ satisfy the star model criterion. We label these transition phases under "star-domainness" in the figure. This observation further confirms our theoretical result.

**Setup.** We consider two-layer linear networks trained using binary cross-entropy loss. We use a 2D binary classification dataset consisting of samples from class-conditional Gaussians. Specifically, $\mathcal{D} = (x_i, y_i); x_i \sim \mathcal{N}(\mu_{y_i}, \Sigma), x \in \mathbb{R}^2, y \in 0, 1$. We train $|Z| = 100$ copies of this model, using $100$ random seeds. Next, we fix each model $\phi \in Z$ and compute its barriers with each of the other models $\theta_i \in Z \setminus \phi,$. We define an empirical star model $\theta^\star \in Z$ as a model that has its loss barriers with all the other models in $Z$ bounded below $\epsilon$: $B(\theta^*, \underset{\theta_i \to \theta^*}{\pi}(\theta_i)) < \epsilon$ for all $\theta_i \in Z \setminus \theta^*$. We test networks with widths $m = 2, 4, 8, 32$. Our focus lies on the proportion of empirical star models with a tolerance $\epsilon$ as $m$ increases. An increasing trend in this proportion w.r.t. increasing $m$ would confirm our theoretical result.

**Results.** For each width $m$, we report the proportion (out of 100) of empirical star models, as defined above. We observe that the number of star models at a given tolerance increases as $m$ increases (e.g., at $\epsilon = 10^{-6}$, there are no star models for $m = 2$ while 21 out of 100 models are "star" at $m = 8$). At $m = 32, \epsilon = 10^{-3}$, all models become star models, i.e., convexity is achieved. We present the quantitative results in Table 3.

|  | $\epsilon = 10^{-6}$ | $\epsilon = 10^{-5}$ | $\epsilon = 10^{-3}$ | $\epsilon = 10^{-1}$ |
|---|---|---|---|---|
| $m = 2$ | 0.00 | 0.01 | 0.05 | 0.12 |
| $m = 4$ | 0.03 | 0.08 | 0.25 | 0.40 |
| $m = 8$ | 0.21 | 0.38 | 0.60 | 0.72 |
| $m = 32$ | 0.86 | 0.97 | 1.00 | 1.00 |

Table 3: **Verifying Theorem 2**. We fix different thresholds $\epsilon$ and at each given width, compute the proportion of star models at a tolerance of $\epsilon$. Given the same $\epsilon$, we observe that this proportion increases with the width $m$.

**Conclusion.** We confirm through our experiments that as the network width $m$ increases, the number of star models in the solution set increases.

### A.2.3 VERIFICATION FOR NON-LINEAR NETWORKS

Here, we present additional results in support of Theorem 2, for non-linear two-layer neural networks. We describe them below.

**Motivation and setup.** Our aim is to verify that the proportion of star models in the solution set increases as network width increases. The experimental setup is similar to that in Appendix A.2.2, where we train two-layer neural networks on the task of binary classification. The key difference this time is that we introduce a non-linearity (ELU) between the two layers. At each network width $m$, we train 100 models and for each of these models, we measure the maximum barrier upon linear interpolation with any other model.

**Results and conclusion.** We present our results in Table 4. We notice that the proportion of star models at any given width and tolerance is lower compared to the completely linear case; however, this proportion still increases (for example, at tolerance $\epsilon = 1e - 6$, it goes from 0.00 at $m = 2$ to 0.93 for $m = 32$) as width increases. Once again, we confirm that our overall conclusion stands consistent with the claim in Theorem 1, even after introducing a non-linearity between the layers.

|  | $\epsilon = 10^{-6}$ | $\epsilon = 10^{-5}$ | $\epsilon = 10^{-3}$ | $\epsilon = 10^{-1}$ |
|---|---|---|---|---|
| $m = 2$ | 0.00 | 0.00 | 0.00 | 0.00 |
| $m = 4$ | 0.00 | 0.00 | 0.00 | 0.04 |
| $m = 8$ | 0.00 | 0.00 | 0.00 | 0.71 |
| $m = 32$ | 0.93 | 1.00 | 1.00 | 1.00 |

Table 4: **Verifying Theorem 2 for non-linear networks**. We fix different thresholds $\epsilon$ and at each given width, compute the proportion of star models at a tolerance of $\epsilon$. Given the same $\epsilon$, we observe that this proportion increases with the width $m$.

### A.3 DIRECTIONS FOR FUTURE WORK

### A.3.1 EXTENSION OF THEORETICAL RESULTS

Our theoretical result in Appendix A.1 is established for a two-layer linear network under simplifying assumptions (Assumption 2). The key ingredient in the proof is the existence of a closed form characterization of the solution set, i.e., $\theta_1 \cdot \theta_2 = C$. Extending this result to more complex cases and proving that star-shaped connectivity arises at much smaller widths than convexity (as shown in our empirical results) would be interesting.

However, deriving theoretical results for full-fledged neural networks is notoriously difficult and, in many cases, infeasible due to their complexity. Theorem 2, proven for the simple yet abundantly challenging case of a 2-layer linear network represents our best attempt at providing a theoretical foundation for our findings.

The most closely related work to our paper is the highly influential work by Entezari et al. (2021), who also tackled the simplified setup of a 2-layer neural network with a ReLU activation in between (Theorem 3.1 in their paper). However, they approach the problem differently and consider random initializations within hypercubes, instead of solutions to a learning problem. Thus, their proof avoids the problem of characterising the solution set of a two-layer NN that contains a non-linearity between the layers. This underscores the broader challenge of developing theoretical proofs related to DNN solution sets. In our work, we take a different route by focusing on actual solutions to an optimization problem, although this necessitates dropping the non-linearity. We believe this puts the complexity of our theoretical results on a comparable footing with Entezari et al. (2021) while still offering a different perspective.

To further address the gap in theoretical results, we provide extensive empirical evidence covering three different architectures, three datasets including ImageNet, and ablation studies in Appendix C

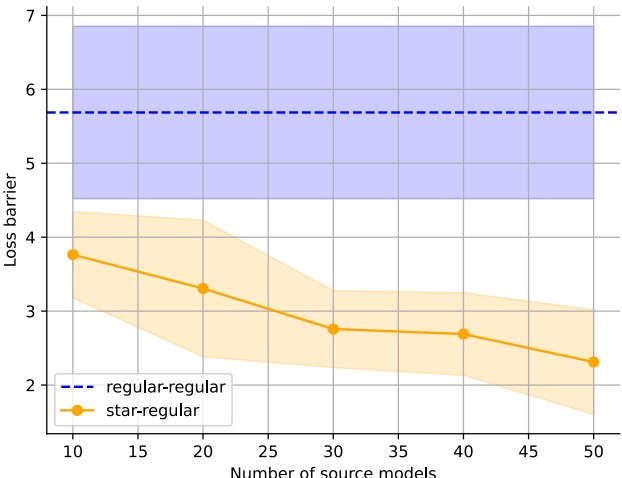

Figure 6: **Transformer results**. Star-regular barrier steadily decreases as we use more source models to train the star model.

and Appendix D. Our experiments indicate significant evidence towards our theoretical result even when the restrictions of Assumption 2 are lifted. We hope that these observations serve as inspiration for other researchers to seek out more rigorous theoretical verification in future work.

### A.3.2 EXTENSION OF EMPIRICAL RESULTS: TRANSFORMERS

As discussed in Appendix A.3.1, our study provides extensive empirical results. However, there is always room to include other settings. For example, transformers exhibit unique permutation symmetries: the latent symmetries within each attention head and the order of the heads themselves. However, there is limited precedent in the literature for empirically investigating transformer-specific permutation symmetries. Prominent studies, including Entezari et al. (2021); Ainsworth et al. (2022); Tatro et al. (2020); Singh & Jaggi (2020), largely exclude transformers from their analyses. Given this, transformers lie outside our primary scope in this study. Considering the current practical relevance of transformers, we believe a dedicated study on the star-domainness of transformer loss landscapes is a promising direction for future research.

In the current work, we instead present small-scale results using a minimal transformer architecture. Since the main focus of our work is star-domainness rather than exploring permutation symmetries in transformers, we conduct our experiments **without permutations**.

**Setup.** We use a minimal ViT model with the MNIST dataset. We train $50$ source models and hold out $3$ additional models for barrier evaluation. Each regular model is trained using the AdamW (Loshchilov & Hutter, 2019) optimizer with a learning rate of $1e-3$ (weight decay $5e-3$) for $100$ epochs. Next, we train star models using more and more source models ($10, 20, 30, 40$ and $50$). Finally, we compare the regular-regular barriers with the star-regular barriers.

**Results and conclusion.** We present results in Table 5 and Figure 6. Our observation here remains consistent with that in the main paper: the star-regular barrier decreases as we increase the number of source models, and the final star-regular barrier of $2.312$ is significantly lower than the regular-regular reference point of $5.687$. This preliminary experiment thus confirms evidence for the star domain conjecture in vision transformers.

**Possible future work.** After completing this study, we encountered the recent work of Verma & Elbayad (2024), which investigates transformer loss landscapes using a data-driven permutation algorithm. Adapting their method to our Algorithm 1 could yield valuable insights. We leave this direction to future work.

Table 5: **Transformer results.** The star model was trained using 50 source models. Our results stay consistent with those in the main paper (Table 1).

| Regular loss | Star loss | Regular-regular | Star-regular |
|---|---|---|---|
| $0.001 \pm 0.001$ | $0.099$ | $5.687 \pm 1.167$ | $2.312 \pm 0.708$ |

## B  IMPLEMENTATION DETAILS

In this section, we describe the setup for replicating our experimental results.

### B.1  MODEL TRAINING

Our model training hyperparameters largely reflect standard practices, but we describe them here for completeness. We used NVIDIA A100 GPUs for most of our experiments. All experiments were performed on single GPUs.

**ResNet18 on CIFAR.**  For ResNet18 models trained on CIFAR-10 and CIFAR-100, we use a batch size of 128. We normalize the data using ImageNet statistics. For data augmentation, we apply padding to the image or its horizontal mirror, and then randomly crop out a $32 \times 32$ region. We train for 200 epochs using SGD with momentum 0.9 and a weight decay of $5e-4$. The initial learning rate is 0.1 and follows a cosine decay schedule to reach 0 by the end of training. Star models and regular models are trained using otherwise identical hyperparameters, except that the star models use the training objective described in Algorithm 1. The differences between different models in $Z$ and $H$ come from the random seed set at the beginning of the training process. We use the following implementation for the ResNet: https://github.com/kuangliu/pytorch-cifar/blob/master/models/resnet.py. Each regular model took roughly 30 minutes to train, while the star model ($|Z| = 50$) took roughly 6 hours to train.

**DenseNet-40-12 on CIFAR.**  Our DenseNet models use largely the same training settings as ResNet18. We highlight the differences here. DenseNet uses a batch size of 64. The weight decay factor is $1e - 4$, and the models are trained for 300 epochs. The learning rate, initially 0.1, is multiplied with 0.1 at epochs 150 and 225. We use the following implementation: https://github.com/andreasveit/densenet-pytorch/blob/master/densenet.py. Star models follow the same training recipe. Each regular model was trained for roughly 3.5 hours. Training the star model took approximately 7 hours.

**VGGs on CIFAR.**  The initial learning rate is set to 0.05 and is multiplied by 0.1 at epochs 100 and 150. Other settings are identical to those used for ResNet18. We use the following implementation: https://github.com/fagp/sinkhorn-rebasin/blob/main/examples/models/vgg.py. Star models follow the same recipe as regular models. It took roughly 15 minutes to train each source model, and 35 minutes to train a star model.

**ResNets on ImageNet.**  For ImageNet, we use a batch size of 256. Models are trained for 100 epochs, using SGD with a learning rate of 0.1 which is multiplied by 0.1 at epochs $30, 60, 90$. The weight decay factor is $1e - 4$. We use the ResNet18 implementation included in PyTorch (Paszke et al., 2019). We leverage the open-source library FFCV (Leclerc et al., 2023) to speed up our experiments. For data augmentation, we resize the image or its horizontal mirror to $256 \times 256$ and randomly crop out a $224 \times 224$ region. Each source model took roughly 13 hours to train, while a star model required about 2 days.

**Weight matching.**  We use weight matching (WM) (Ainsworth et al., 2022). Our implementation leverages an open-source Python package called "rebasin": https://pypi.org/project/rebasin/.

**Total compute.**  We estimate to have spent approximately $50 - 70$ days of NVIDIA A100 compute hours for the experiments in this paper (not including experiments that did not make it into the paper).

## B.2 CALCULATION OF LOSS BARRIERS

We use the definition in **?** to calculate loss barriers between any given pair of models (eq. (1)). Since the definition in eq. (1) involves an infinite search space for the maximum, we sample a finite set $T = \{t_1, t_2, \cdots, t_K\}$ of equi-spaced points and compute the maximum as

$$\max_{t \in T} \mathcal{L}\left((1-t) \cdot \theta_A + t \cdot \theta_B\right) - \left((1-t) \cdot \mathcal{L}\left(\theta_A\right) + t \cdot \mathcal{L}\left(\theta_B\right)\right) \tag{10}$$

Our sampling of equidistant points is consistent with prior work Ainsworth et al. (2022); Guerrero Peña et al. (2023). The size of $T$ itself varies in prior work. Because of the scale of our experiments, we use $|T| = 11$, including the end-points. In Appendix C.1, we show that this size is sufficient for obtaining statistically significant results.

## B.3 HANDLING BATCH NORMALIZATION

Batch normalization (Ioffe & Szegedy, 2015) is integral to efficient DNN training. **?** describe the so-called "variance collapse" problem that leads to degradation of interpolated models. As a solution, we follow Ainsworth et al. (2022) and recalculate the batch statistics for each interpolated model, by performing one forward pass through the entire training set before performing evaluation.

## B.4 NOTE ON DEFINING SOLUTIONS

In general, a "solution" in this paper refers to the global minima of the objective function. Below, we justify this definition.

**Theory.** In Theorem 1, we ensure global minima by assuming precisely zero loss. In Conjecture 2, we also assume global minima that are theoretically reachable by SGD.

**Practice.** In experiments involving DNNs, it is tricky to determine whether global minima were reached. However, we offer the following assurances.

- For our **CIFAR10 and CIFAR100 experiments**, we reach extremely low training loss values (for example, training loss in the order of $10^{-3}$ for CIFAR10-ResNet18 models). These results strongly suggest that the obtained models are very close to the global minima of the objective function.

- For our **ImageNet experiments**, the loss values are not exactly zero. However, the loss at the actual global minimum of the ImageNet classification task is an undetermined quantity because of inherent label noise. Nonetheless, our models reach state-of-the-art accuracy for the given dataset and architecture ($> 70\%$ validation accuracy for ResNet18). Given this, we believe that our models serve as reasonable and sufficiently reliable proxies for global minima.

## C STATISTICAL SIGNIFICANCE OF OUR RESULTS

In this section, we validate our choices concerning the reporting of our results and demonstrate that our findings are statistically significant.

## C.1 SAMPLE SIZES FOR INTERPOLATION

An essential part of our experimental setup involves computing the loss barrier between two given networks. To achieve this, we selected a set of equally spaced points between $t = 0$ and $t = 1$ and evaluated the interpolated models at these points. This process is computationally intensive and becomes slower as the number of interpolation points increases. Throughout this study, we used the points $t = 0.0, 0.1, \ldots, 1.0$. In this section, we demonstrate that the number of interpolation points we used does not negatively impact the significance of our results. To this end, we conducted an ablation study on CIFAR10-ResNet18 models, varying the number of interpolation points for

computing the loss barrier. We present the comparison in Table 7. As we increased the number of interpolation points from 11 to 51, we observed a decrease of 0.007 in the "regular-regular" barrier and an increase of 0.004 in the "star-regular" barrier. For reference, these differences are less than the standard deviations in the corresponding observations, and are thus statistically insignificant for our final conclusions.

## C.2 SIZE OF HELD-OUT SET

To reduce noise in our results, we compare barriers after computing them for several model pairs. The size of the held-out set $|H|$ is usually 5, and sometimes even 3. Here, we confirm that this is a large enough sample size and that considering a larger set of held-out models does not change our results fundamentally. In particular, we vary the number of held-out models $|H|$ and source models $|Z|$ and obtain the corresponding mean barrier values as well as standard deviations. First, we set both $|H|$ and $|Z|$ to 3. Then, we set $|H| = |Z| = 5$ and finally, $|H| = |Z| = 15$. In each case, we interpolate all held-out models with all source models. Hence, in the last case, we perform 225 "regular-regular" barrier computations and 15 "star-regular" barrier computations. We present the results in Table 8. We observe that the average "regular-regular" loss barrier between two arbitrary models remains larger than 0.37 throughout, with a standard deviation close to 0.05. In contrast, the average "star-regular" barrier remains lower than 0.1, with a standard deviation lower than 0.02. None of the observed metrics or our conclusions change significantly when increasing the number of samples. This observation provides confidence that our practice of setting the number of held-out models to 3 or 5 provides reliable estimates while also being computationally cheaper.

## C.3 MAXIMUM AND MINIMUM BARRIERS

Throughout the study, we consider mean values of "regular-regular" and "star-regular" loss barriers for comparison. Here, we additionally compare maximum and minimum barrier values for each model pair and confirm that the same trend holds, *i.e.,* "star-regular" barriers are lower than "regular-regular" barriers. We present the results in Table 8. We observe that as the sample size increases, the minimum barrier values go down, while the maximum barrier values go up for both "regular-regular" and "star-regular" pairs. But the minimum barrier obtained by "regular-regular" pairs is still 0.25, which is significantly higher than even the maximum "star-regular" barrier, *i.e.,* 0.117. This observation confirms that the star model, on average, exhibits better linear connectivity with other arbitrary models, than even the most "connected" arbitrary models exhibit between each other.

# D ABLATIONS

## D.1 TEST METRICS

The main discourse around mode connectivity (Garipov et al., 2018) as well as convexity (**?**) is built around training loss and accuracy. Therefore, our empirical observations in support of the star domain conjecture also primarily use the training set. However, applications like Bayesian model averaging (Section 4.1) require our conclusions to hold for the test set. Therefore, we also examine the veracity of our claims with respect to test loss and accuracy. We present our representative findings in Table 6. Similarly to the training set results (Table 1), we observe that the "star-regular" test loss barriers are consistently lower than "regular-regular" test loss barriers. Notably, for VGG11, our star model achieves a zero test loss barrier with regular models, in comparison to a barrier of 0.24 between two regular models. Overall, our observations indicate that the star domain conjecture holds for both training and test losses.

## D.2 CONNECTIVITY TO ADAM SOLUTIONS

Throughout the study, we have largely focused on the SGD-trained solutions and the star model built from SGD-trained source models. Here, we examine the connectivity between SGD solutions, SGD-induced star models and Adam-trained solutions. Figure 7 shows the loss landscape across different types of solutions. We observe, as before, that the loss barrier between SGD solutions and our star model $\theta^\star$ is nearly non-existent, while the SGD solutions are generally not linearly connected. Our star model $\theta^\star$ shows less connectivity with Adam solutions (the curve between "adam" and

Table 6: **Empirically verifying the star domain conjecture.** "Regular loss" and "Star loss" indicate *test* losses for regular models in $Z$ and star models $\theta^\star$, respectively. "Star-regular" refers to the barrier $B(\theta^\star, \theta_h)$ between a star model and one of the heldout models in $H$. For comparison, "Regular-regular" is the *test* loss barrier $B(\theta_A, \theta_B)$ between two arbitrary models. We report values up to one standard deviation over several runs, except for ImageNet. In each case, star models exhibit significantly lower loss barriers with other models, than the corresponding average loss barrier between two regular models. This trend is consistent with our observation for training losses in Table 1.

| Dataset | Architecture | Regular loss | Star loss | Regular-regular | Star-regular |
|---|---|---|---|---|---|
| CIFAR-10 | ResNet-18 | $0.181 \pm 0.005$ | $0.222$ | $0.336 \pm 0.057$ | $0.035 \pm 0.007$ |
| CIFAR-10 | ResNet-18 (ADAM) | $0.334 \pm 0.010$ | $0.299$ | $1.099 \pm 0.516$ | $0.168 \pm 0.015$ |
| CIFAR-10 | VGG11 | $0.421 \pm 0.011$ | $0.456$ | $0.242 \pm 0.036$ | $0.000 \pm 0.000$ |
| CIFAR-10 | VGG19 | $0.444 \pm 0.019$ | $0.395$ | $0.903 \pm 0.150$ | $0.117 \pm 0.067$ |
| CIFAR-10 | DenseNet | $0.269 \pm 0.012$ | $0.290$ | $4.405 \pm 0.730$ | $1.612 \pm 0.408$ |
| CIFAR-100 | ResNet-18 | $0.925 \pm 0.007$ | $1.216$ | $2.306 \pm 0.039$ | $0.447 \pm 0.051$ |
| CIFAR-100 | DenseNet | $1.312 \pm 0.019$ | $1.115$ | $5.613 \pm 0.209$ | $3.005 \pm 0.176$ |
| ImageNet-1k | ResNet-18 | $1.203$ | $1.634$ | $5.477$ | $2.548$ |

Table 7: **Different sample sizes for computing loss barriers.** We report the "regular-regular" and "star-regular" loss barriers computed using eq. (10), with different sizes of the set $T$ of interpolation points. Using 51 interpolation points instead of 11 does not lead to a significant change in the computed loss barriers.

| Sample size | Regular-regular | Star-regular |
|---|---|---|
| 11 | $0.376 \pm 0.055$ | $0.090 \pm 0.007$ |
| 21 | $0.369 \pm 0.053$ | $0.089 \pm 0.006$ |
| 51 | $0.369 \pm 0.053$ | $0.094 \pm 0.008$ |

Table 8: **Computing average loss barriers over different sizes of the sets of source models $Z$ and heldout models $H$.** We compute minimum, average and maximum "star-regular" barriers ("SR-min", "SR-avg." and "SR-max." respectively) over varying numbers of heldout models $H$. For comparison, we compute minimum, average and maximum "regular-regular" barriers ("RR-min", "RR-avg." and "RR-max." respectively). Increasing the number of held-out models from 3 to 15 does not significantly change the observed trend.

| Sample size | RR-min. | SR-min. | RR-avg. | SR-avg. | RR-max. | SR-max. |
|---|---|---|---|---|---|---|
| 3 | 0.300 | 0.082 | $0.376 \pm 0.055$ | $0.090 \pm 0.007$ | 0.445 | 0.095 |
| 5 | 0.292 | 0.069 | $0.381 \pm 0.059$ | $0.077 \pm 0.007$ | 0.530 | 0.084 |
| 15 | 0.255 | 0.071 | $0.382 \pm 0.046$ | $0.093 \pm 0.014$ | 0.540 | 0.117 |

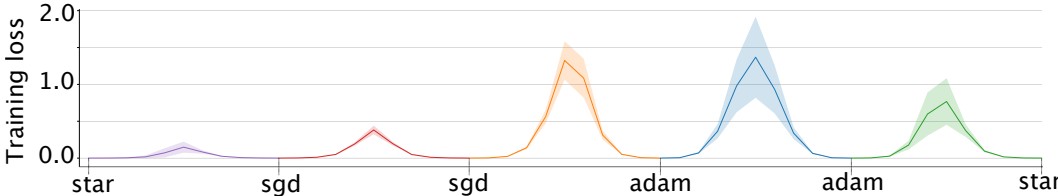

Figure 7: **Training loss landscape across SGD models, Adam models, and SGD-induced star models.** We plot test loss across different types of solutions in $S$. Our star model $\theta^\star$ ("star" in the plot) is constructed from a set of SGD-trained models $Z$. We note that the star model is well-connected with SGD solutions. There remains a loss barrier between the star model and Adam solutions, but it is significantly lower than the barrier among Adam solutions.

"star-sgd") than with SGD solutions. However, we note that the loss barrier is significantly lower than for the linear interpolations between pairs of Adam solutions (the curve between "adam" and "adam"). Based on this observation, we conclude that while the scope of our conjecture remains within SGD-trained solutions, there are hints that our star model shows enhanced connectivity with other types of solution subsets.

### D.3 COMPARISON WITH SINKHORN-REBASIN

Guerrero Peña et al. (2023) introduced a novel permutation-finding algorithm *viz., Sinkhorn-rebasin* aimed at reducing the loss barriers between two arbitrary models. When using the data-free setting, the method Guerrero Peña et al. (2023) can be considered a differentiable form of weight matching Ainsworth et al. (2022). The authors notably demonstrate that their method performs better than weight matching on average, albeit it only partially eliminates loss barriers between two given models. However, Sinkhorn-rebasin does not yet support networks with skip connections, making it unsuitable for our experiments involving ResNets and DenseNets. Additionally, Sinkhorn-rebasin requires hyperparameter tuning (such as optimizer and learning rate), which could introduce confounding factors into our experiments.

The primary objective of this paper is to investigate the empirical validity of our star domain conjecture. We find weight matching sufficient for this purpose. However, our empirical verification is based upon comparing "regular-regular" barriers, *i.e.,* loss barriers between two arbitrary solutions, and "star-regular" barriers, *i.e.,* loss barriers between the star model and other arbitrary solutions. it is important to verify how much Sinkhorn-rebasin can further improve these "regular-regular" loss barriers. To this end, we compare "star-regular" barriers "regular-regular" barriers after applying Sinkhorn-rebasin (SH) instead of Weight Matching. We use VGG19 models with batch normalization, trained on CIFAR-10, for this purpose. First, we perform a hyperparameter search on the learning rate for the permutation-finding algorithm Guerrero Peña et al. (2023), using $\mathcal{C}_{L2}$ distance (as described in Guerrero Peña et al. (2023)) as the optimization objective. Our search space is the set $0.01, 0.1, 1.0, 10.0, 20.0, 30.0, 40.0, 50.0, 60.0, 80.0, 100.0, 120.0, 150.0, 200.0$. We observe the best "regular-regular" loss barrier between two pretrained models from Guerrero Peña et al. (2023), at a learning rate of $150.0$. This loss barrier forms the reference point "regular-regular" in our comparison. Next, we train a star model using our own source models, and then compute the "star-regular" barrier with one of the pre-trained models from Guerrero Peña et al. (2023). The results are presented in Figure 8.

The "regular-regular" barriers obtained in our experiment are comparable to those reported in Figure 6 of Guerrero Peña et al. (2023). We observe that in this particular case, Sinkhorn-rebasin exhibits higher loss barriers than weight matching, although it might be possible to reduce this barrier further with a different set of hyperparameters. Nevertheless, "star-regular" barriers remain lower than "regular-regular" barriers in both cases. Future work may look more closely into the effect of using different permutation algorithms.

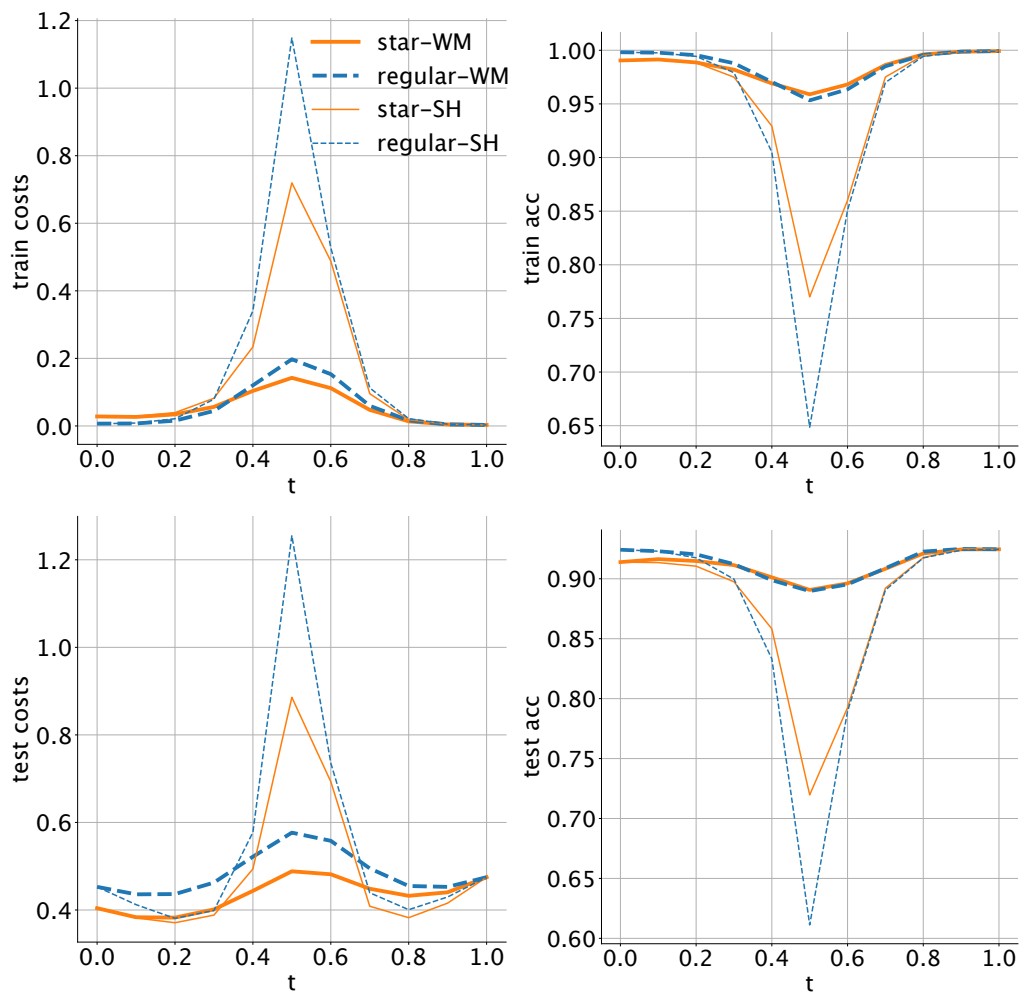

Figure 8: **Effect of using Sinkhorn-rebasin instead of weight matching.** We compare "star-regular" barriers against "regular-regular" barriers after fixing the permutation algorithm used to match the weights of the two models being interpolated. star-WM and regular-WM refer to the barriers after applying weight matching Ainsworth et al. (2022). Similarly, star-SH and regular-SH refer to barriers after applying Sinkhorn-rebasin Guerrero Peña et al. (2023). While vanilla WM outperforms SH in this case, a better hyperparameter choice may eventually cause SH to outperform WM. Nevertheless, the findings in this investigation further support our conclusion that star models are well-connected to other regular models, in comparison to how well the regular models are connected amongst themselves.

## D.4 DIFFERENT SAMPLING SCHEMES FOR MONTE-CARLO PPTIMIZATION

While training our star models using Starlight (Algorithm 1), we sample the interpolation factor $t$ from $\mathrm{Unif}[0, 1]$. However, empirical results show that loss barriers are typically achieved close to the center of the interpolation line ($t = 0.5$). This raises the question: would Starlight be more efficient if the sampling scheme placed more weight toward the center of the interpolation line? To test this, we run ablations with different sampling schemes.

- **Uniform (used in main paper)**: $t \sim \mathrm{Unif}[0, 1]$.

- Beta: $t \sim \mathrm{Beta}(2, 2)$. We sample more around $t = 0.5$.

- Constant 0.5: $t = 0.5$.

Table 9: **Different sampling schemes for Starlight.** We report the "star-regular" and "regular-regular" barriers for each sampling case; averages were computed over three different runs using different random seeds. While sampling from a beta distribution "Beta" performs slightly better, the difference is too small to be statistically significant.

| Sampling scheme | Regular loss | Star loss | Regular-regular | Star-regular |
|---|---|---|---|---|
| Unif$[0, 1]$ | $0.001 \pm 0.000$ | $0.001 \pm 0.000$ | $0.383 \pm 0.056$ | $0.084 \pm 0.012$ |
| Beta$(2, 2)$ | – do – | $0.001 \pm 0.000$ | – do – | $0.069 \pm 0.010$ |
| Constant$0.5$ | – do – | $0.082 \pm 0.004$ | – do – | $0.018 \pm 0.009$ |

We show the results in Table 9. We observe that both the Uniform and Beta sampling schemes obtain star models with identical training loss (0.001). Beta achieves a slightly better loss barrier (0.069) than Uniform (0.084), while the difference is not particularly significant in this case. Constant sampling obtains a much worse star model in terms of training loss (0.08). However, the finding with Beta suggests that Starlight can potentially benefit from better sampling schemes for $t$ in future work.

### D.5 EFFECT OF PERTURBATIONS

While the main point of our work is to show that *a* star model exists, it would also be interesting to find out whether other models in the vicinity of the discovered star model, also exhibit star-model-like properties. To test this, we slightly perturbed the CIFAR10-ResNet18 star model and evaluated its "starness". Specifically, we sampled points $\widetilde{\theta}_\epsilon = \theta^* + \epsilon \cdot r$ from the surface of a sphere of radius $\epsilon$ around $\theta^*$. We consider three samples for each perturbation radius, and calculate barriers with three held-out models. Results can be found in Table 10. Indeed, perturbations with $\epsilon = 16$ are still good star model candidates.

Table 10: **Barriers after perturbation.** Even after perturbation, star-regular barriers remain substantially lower than the regular-regular barrier of $0.38$, indicating that there exist several star models instead of just one.

| $\epsilon$ | Star loss | Star-regular barrier |
|---|---|---|
| $2^1$ | $0.001 \pm 0.000$ | $0.078 \pm 0.009$ |
| $2^2$ | $0.001 \pm 0.000$ | $0.080 \pm 0.009$ |
| $2^3$ | $0.002 \pm 0.000$ | $0.086 \pm 0.007$ |
| $2^4$ | $0.012 \pm 0.001$ | $0.120 \pm 0.013$ |

### D.6 MODIFIED STARLIGHT: STAR-REGULAR BARRIERS

Here, we investigate the starness of star models trained with the modified objective in Section 4.2. We consider ResNet18 models trained on CIFAR-10 and CIFAR-100, reporting the mean of $3 \times 3 = 9$ star-regular barriers for each case. Results can be found in Table 11. Similar to the trend shown in Figure 1, the star-regular barriers steadily decrease (e.g., from $0.31$ to $0.12$ for CIFAR-10) with an increasing number of source models, and remain significantly lower than the average regular-regular barrier (e.g., $0.38$ for CIFAR-10).

Table 11: **Star model barriers with the modified objective, after the addition of a cross-entropy term to the loss**. The barriers follow the decreasing trend shown in Figure 1, and for 50 source models, they become significantly lower than the average regular-regular barrier.

| Dataset | #source | Star-regular barrier |
|---------|---------|----------------------|
| CIFAR-10 | 2 | $0.31 \pm 0.05$ |
| CIFAR-10 | 5 | $0.25 \pm 0.05$ |
| CIFAR-10 | 50 | $0.12 \pm 0.02$ |
| CIFAR-100 | 2 | $2.70 \pm 0.06$ |
| CIFAR-100 | 5 | $2.10 \pm 0.10$ |
| CIFAR-100 | 50 | $0.76 \pm 0.09$ |

### D.7 ROLE OF ACTIVATION FUNCTION

Our experiments largely make use of networks with the ReLU activation, a homogeneous function. To test the influence of ReLU's homogeneity on our results, we conducted minimal experiments using two non-homogeneous activation functions: ELU (Clevert et al., 2016) and Swish (Ramachandran et al., 2018). Results are presented in Table 12. Even in this minimal setup, we observe that star-regular barriers are significantly lower than regular-regular barriers (for instance, $0.224$ vs $0.497$ for ELU). We expect the star-regular barriers to get better with more source models. Our results indicate that star-domainness can emerge even for non-homogeneous activation functions.

Table 12: **Barriers with different activation functions.** We used 3 held-out models and 10 source models in each case. We observe lower star-regular than regular-regular barriers even in this minimal setup using non-homogeneous activation functions. This indicates that "starness" may not be a consequence of the homogeneous property of the function. As a reference point, we report the star-regular barrier for ReLU star models trained using 10 source models.

| Activation function | Regular loss | Star loss | Regular-regular | Star-regular |
|---------------------|--------------|-----------|-----------------|--------------|
| ReLU | $0.001 \pm 0.001$ | $\approx 0.001$ | $0.383 \pm 0.056$ | $\approx 0.19$ |
| ELU | $0.007 \pm 0.001$ | $0.068$ | $0.497 \pm 0.174$ | $0.224 \pm 0.021$ |
| Swish (SiLU) | $0.001 \pm 0.000$ | $0.009$ | $0.347 \pm 0.046$ | $0.208 \pm 0.019$ |

## E ADDITIONAL RESULTS

Our main results are reported in Figure 2 and Table 1, in the main paper. For the sake of completeness, we report interpolation plots for the rest of our experiments in this section. We include plots for DenseNet (Figure 9), VGG (Figure 10), and ImageNet-1k (Figure 11).

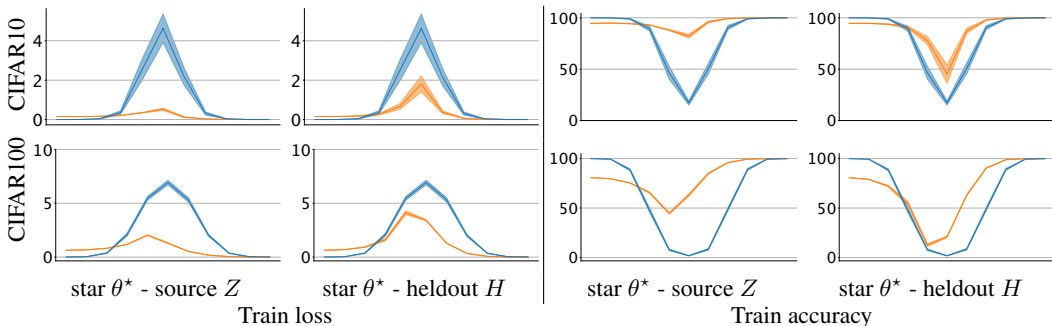

Figure 9: **Loss barriers for DenseNet-40-12 star models on CIFAR**. We plot training loss and accuracy curves obtained upon interpolation between star-regular and regular-regular models pairs. Star-regular loss barriers continue to be lower than regular-regular barriers, as observed in Figure 2.

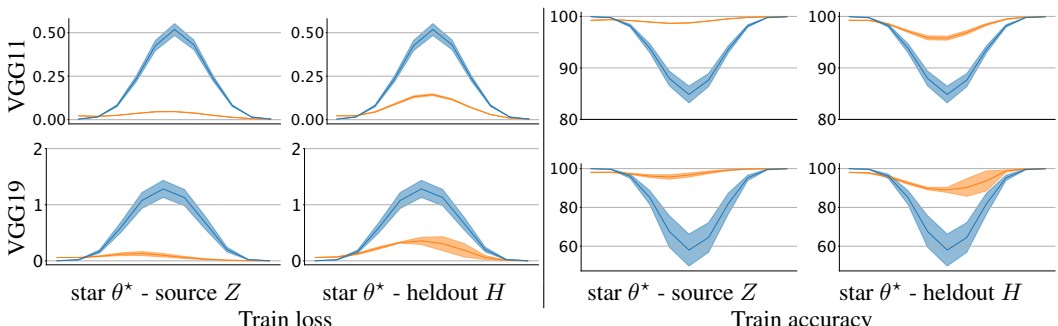

Figure 10: **Loss barriers for VGG star models trained on CIFAR-10**. Training loss and accuracy curves obtained upon interpolation between star-regular and regular-regular model pairs. We observe the same trend as in Figure 2.

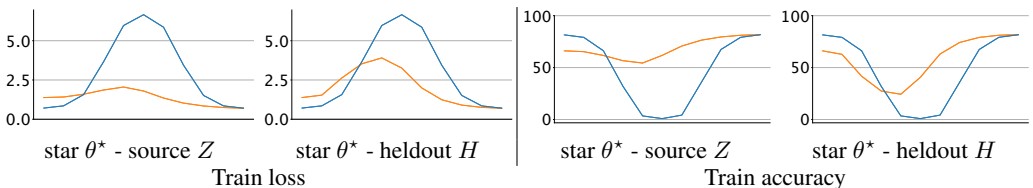

Figure 11: **Loss barriers for ResNet18 star models trained on ImageNet-1k**: training loss and accuracy curves obtained upon interpolation between star-regular and regular-regular model pairs. While ImageNet models struggle to achieve LMC, star-regular barriers still fare much better than regular-regular barriers.

