# OpenReview forum: "Do Deep Neural Network Solutions Form a Star Domain?"
_ICLR.cc/2025/Conference — ICLR 2025 Poster_

### Official Review · Reviewer_gkBc · 2024-10-21

**Soundness:** 3
**Presentation:** 3
**Contribution:** 3
**Rating:** 6
**Confidence:** 3

**Summary:**

The paper presents a conjecture regarding the nature of solution sets for deep neural networks (DNNs) trained via stochastic gradient descent (SGD). It introduces the concept of a "star domain," proposing that these solution sets can be characterized by a special "star model" that connects to all other solutions in a low-loss path, modulo permutation invariances. The paper proposes the ``Starlight" algorithm that finds a star model of a given learning task and confirms existing reports that the convexity conjecture requires very wide networks to hold based on experimental results.

**Strengths:**

The introduction of the star domain conjecture adds a novel perspective to existing theories regarding the connectivity of neural network solutions. The Starlight algorithm is well-defined and offers practical means to identify star models, contributing to the field of model ensemble methods. The paper includes thorough empirical validation across various architectures and datasets, strengthening the claims made.

**Weaknesses:**

While the empirical results are promising, the theoretical underpinnings of the star domain conjecture could be elaborated further. The authors acknowledge challenges in validating the conjecture theoretically but could provide more insights into potential avenues for future theoretical exploration.

The experiment results primarily focus on specific settings (e.g., ResNet, CIFAR-10). Broader validation across different tasks and architectures could enhance the generalizability of the findings.

The algorithm's complexity may pose challenges for reproducibility. Detailed discussions on implementation and computational overhead would be beneficial.

**Questions:**

-  Theorem 1 is established for an oversimplified two-layer linear network. Can the author at least show results with two-layer network with nonlinear activations? Otherwise, the theory is too weak to support the star domain conjecture.

-  Follow-up questions : How do the authors envision extending the theoretical framework of the star domain conjecture to encompass a broader range of neural network architectures and training methods? What specific conditions or assumptions must hold for the star domain conjecture to be valid in practical applications?

-  About Empirical Validation: how do the authors plan to validate the star domain conjecture across different types of neural network architectures (e.g., transformers, recurrent networks)? Can the authors provide additional insights or results regarding the performance of the Starlight algorithm in high-dimensional parameter spaces?

- About algorithm complexity: Given the complexity of the Starlight algorithm, what steps have the authors taken to ensure its computational efficiency and ease of implementation in practical settings? Are there potential optimizations or modifications to the Starlight algorithm that could enhance its performance, especially in large-scale applications?

---

> ### Author Response · Authors · 2024-11-22
> **(1/3) Addressing Reviewer gkBc's concerns: Weaknesses**
>
> Dear Reviewer gkBc,
>
> We are grateful for your thoughtful evaluation of our work.
>
> **Weakness 1: "The theoretical underpinnings of the star domain conjecture could be elaborated further. The authors acknowledge challenges in validating the conjecture theoretically but could provide more insights into potential avenues for future theoretical exploration."**
>
> We validate our conjecture for two-layer linear networks (Appendix A). The immediate next step in extending our current theoretical results would be to introduce a non-linear activation between the two layers.
>
> However, the current proof relies on the fact that $\theta^* = \theta_1\cdot\theta_2$ is a linear mapping and has a single correct solution. This allows us to sufficiently characterize the solution set of the two-layer linear network.
>
> To introduce a non-linearity into our theoretical framework, we would need to characterize the solution set for 2-layer *non-linear* neural nets instead. This characterization problem is non-trivial and still remains elusive in the DNN theoretical research community.
>
> Extension **even beyond 2-layer NNs** is currently very difficult. There is hardly any existing theory on loss landscapes and solution sets for non-linear neural networks with more than 2 layers.
>
> **Weakness 2: "The experiment results primarily focus on specific settings (e.g., ResNet, CIFAR-10). Broader validation across different tasks and architectures could enhance the generalizability of the findings."**
>
> We agree that broader empirical validation can be valuable. Currently, our empirical results comprise:
>
> - **Datasets:** CIFAR-10, CIFAR-100 and ImageNet (Table 1);
> - **Architectures:** several variants of ResNets, VGGs, and DenseNets (Table 1);
> - **Network sizes:** ResNets with varying widths and depths (Figure 3, lines 402-413);
> - **Optimizers:**: SGD (most of our experiments) and Adam (lines 415-424, Figure 6, Appendix D.2);
> - **Several ablations on statistical significance** (Appendix C);
> - **Different activation functions** (Appendix D.7);
> - **Different sampling schemes for the MC estimation** (Appendix D.4)
>
> and so on. In the interest of brevity, the above list is not exhaustive. Given the breadth of this empirical analysis, we believe our results provide sufficient coverage to validate our claims. We remain open to further suggestions.
>
> **Weakness 3: "The algorithm's complexity may pose challenges for reproducibility. Detailed discussions on implementation and computational overhead would be beneficial."**
>
> We address the concerns raised below.
>
> **Reproducibility:** We have a code base ready to be shared to public upon acceptance. The readers will be able to run inference and training of the star model with the shared code and weights.
>
> **Implementation**: Algorithm 1 enumerates the details of our approach step-by-step. §3 describes how we address the intractability of solving Eqn. (4) using a Monte Carlo optimization scheme. In Appendix B, we carefully describe our experimental setup, including hyperparameters and configurations.
>
> **Computational overhead:** Most of our experiments were carried out using a single NVIDIA A100 GPU with 40 GB of memory, therefore, they remain reasonably feasible. In order to mitigate computational overhead, we take additional steps such as performing the permutation-finding step only once per epoch (lines 286-303 of the updated paper).
>
> We hope this addresses your concern and are looking forward to answering any further questions about this aspect.

---

> ### Author Response · Authors · 2024-11-22
> **(2/3) Addressing Reviewer gkBc's concerns: Questions 1-2**
>
> **Question 1: "Theorem 1 is established for an oversimplified two-layer linear network. Can the author at least show results with two-layer network with nonlinear activations? Otherwise, the theory is too weak to support the star domain conjecture."**
>
> Thank you for the thoughtful suggestion. We have now obtained additional results in support of Theorem 1, for non-linear two-layer neural networks. We describe them below.
>
> **Motivation and setup.** Our aim is to verify that the proportion of star models in the solution set increases as network width increases. The experimental setup is similar to that in Appendix A.2.2, where we train two-layer neural networks on the task of binary classification. The key difference this time is that we introduce a non-linearity (ELU) between the two layers. At each network width $m$, we train $100$ models and for each of these models, we measure the maximum barrier upon linear interpolation with any other model.
>
> **Results and conclusion.** We present our results below. We notice that the proportion of star models at any given width and tolerance is lower compared to the completely linear case; however, this proportion still increases (for example, at tolerance $\epsilon=1e-6$, it goes from $0.00$ at $m=2$ to $0.93$ for $m=32$) as width increases. Once again, we confirm that our overall conclusion stands consistent with the claim in Theorem 1, even after introducing a non-linearity between the layers.
>
> |       | $\epsilon=10^{-6}$ | $\epsilon=10^{-5}$ | $\epsilon=10^{-3}$ |   $\epsilon=10^{-1}$   |
> | ----- | ------------------ | ------------------ | ------------------ | --- |
> | $m=2$ |        $0.00$            |     $0.00$           |     $0.00$         | 0.00    |
> | $m=4$ |         $0.00$           |      $0.00$          |         $0.00$       | 0.04    |
> | $m=8$ |   $0.00$                 |       $0.00$         |   $0.00$             |    0.71 |
> | $m=32$ |     $0.93$               |  $1.00$              |              $1.00$  | 1.00    |
>
> We have included this discussion in Appendix A.2.3 of the updated PDF.
>
> **Question 2.1: "How do the authors envision extending the theoretical framework of the star domain conjecture to encompass a broader range of neural network architectures and training methods?"**
>
> Our current theoretical proof is established for a two-layer linear network. Extending it to include more neural network architectures and training methods is a promising avenue for future work. The first step here would be establishing the result in Theorem 1 for two-layer non-linear networks. However, even this step is currently very challenging because of the lack of literature characterising the solution set for two-layer neural networks containing non-linearities. Please also see our response to Weakness 1 above.
>
> **Question 2.2: "What specific conditions or assumptions must hold for the star domain conjecture to be valid in practical applications?"**
>
> Assuming state-of-the-art neural networks that contain permuation symmetries and were trained using SGD, we expect our results to continue to hold in similar practical setups. This is based on the observation that so far, the star domain conjecture seems to hold for all the architectures (ResNets of various widths and depths, VGGs and DenseNets) and datasets (CIFAR-10, CIFAR-100, and ImageNet) considered in the paper (Table 1) and even under various ablations like a change of optimizer or activation function (Appendix D).

---

> > ### Comment · Reviewer_gkBc · 2024-11-23
> >
> > I thank the authors for their efforts to provide a detailed response. Regarding Theorem 1, I just realized that the author may refer to [1], which computed the number of equivalent parameter manifolds for overparameterized nets when vectors are replicated and describe the geometry (in terms of manifold and connected affine subspaces) of sets of minima and critical points when increasing the width of a network. And relevant references [2,3,4].
> >
> > Regarding the extension to transformers and other types of architecture, the permutation invariance generally exists in attention modules and any network with linear layers. The author can refer to [5,6].
> >
> >
> >
> >
> >
> > [1] Simsek, Berfin, et al. "Geometry of the loss landscape in overparameterized neural networks: Symmetries and invariances." International Conference on Machine Learning. PMLR, 2021.
> > [2] Ainsworth, Samuel K., Jonathan Hayase, and Siddhartha Srinivasa. "Git re-basin: Merging models modulo permutation symmetries." arXiv preprint arXiv:2209.04836 (2022).
> > [3] Entezari, Rahim, et al. "The role of permutation invariance in linear mode connectivity of neural networks." arXiv preprint arXiv:2110.06296 (2021).
> > [4] Jordan, Keller, et al. "Repair: Renormalizing permuted activations for interpolation repair." arXiv preprint arXiv:2211.08403 (2022).
> > [5] Jacot, Arthur, et al. "Saddle-to-saddle dynamics in deep linear networks: Small initialization training, symmetry, and sparsity." arXiv preprint arXiv:2106.15933 (2021).
> > [6] Shen, G. (2023). Exploring the Complexity of Deep Neural Networks through Functional Equivalence. arXiv preprint arXiv:2305.11417.

---

> > > ### Author Response · Authors · 2024-11-27
> > >
> > > Dear Reviewer gkBc,
> > >
> > > Thank you for your continued engagement with our work. We are diligently working on our response to ensure we answer your questions thoroughly. It's taking longer than we thought.
> > >
> > > We will submit our detailed reply and make sure to upload the updated PDF with the implemented changes as well as those proposed by Reviewer 3x6L, before 23:59 on November 27th, AoE.
> > >
> > > Thank you for your understanding and patience.
> > >
> > > Best,
> > >
> > > Authors

---

> ### Author Response · Authors · 2024-11-22
> **(3/3) Addressing Reviewer gkBc's concerns: Questions 3-4**
>
> **Question 3.1: "About Empirical Validation: how do the authors plan to validate the star domain conjecture across different types of neural network architectures (e.g., transformers, recurrent networks)?"**
>
> We show results for all major architecture families that are typically used in related literature (ResNets, VGGs, DenseNets).
>
> Prior work in the domain of mode connectivity modulo permutations has not thoroughly explored permutation-finding algorithms for other kinds of architectures (transformers, recurrent networks, Mamba and ConvNeXt-based architectures).
>
> This is because these architectures by default contain significantly fewer permutation symmetries. For example, Ainsworth et al. \[1\] mention ConvNeXt as a failure mode in Appendix A.1 of their paper, because of "surprisingly few permutation symmetries due to extensive use of depth-wise convolutions". The documentation of the repository that we use for weight matching (mentioned in line 990 of the updated paper) further mentions that "transformer models don't work well" for this same reason.
>
> Given this, we decide not to consider these architectures in the current work.
>
> **Question 3.2: "Can the authors provide additional insights or results regarding the performance of the Starlight algorithm in high-dimensional parameter spaces?"**
>
> Our empirical results were obtained under fairly high-dimensional parameter spaces. For instance, the ResNet models trained on CIFAR-10 consist of $11.17$ million parameters each. The WRN-$22\times8$ models are even larger at $17.16$ million parameters. We would be happy to engage in further discussion and explore additional dimensions upon clarification regarding the specific scale or dimensionality you are interested in investigating.
>
> **Question 4.1: "About algorithm complexity: Given the complexity of the Starlight algorithm, what steps have the authors taken to ensure its computational efficiency and ease of implementation in practical settings?"**
>
> Thank you for raising this concern. §3 describes how we use a Monte Carlo optimization scheme to address the intractability of solving Eqn. (4). We carried out most of our experiments using a single NVIDIA A100 GPU with 40 GB of memory, therefore, we assure you and other readers that our experiments are reasonably feasible. Our answer to Weakness 3 above may also be relevant here.
>
> **Question 4.2: "Are there potential optimizations or modifications to the Starlight algorithm that could enhance its performance, especially in large-scale applications?"**
>
> Our current implementation has been optimized to ensure scalability and efficiency (as detailed in our answer to Weakness 3 above). One potential enhancement to increase scalability is a different sampling approach for source models: instead of training $n$ source models independently, one may try running only $n/k$ independent training runs and sampling the last $k$ checkpoints from each run. We leave this direction for future work to explore. We thank you again for this question and are happy to address any further inquiries.
>
> \[1\] Ainsworth, Samuel, Jonathan Hayase, and Siddhartha Srinivasa. "Git Re-Basin: Merging Models modulo Permutation Symmetries." ICLR 2022.

---

> ### Author Response · Authors · 2024-11-28
> **Addressing the second part of Reviewer gkBc's question (regarding extension to transformers and other architectures)**
>
> We thank the reviewer for their continued engagement with our work.
>
> Permutation symmetries indeed exist in consecutive linear layers [5] and attention modules, where attention heads, as well as the query and key embeddings within each head, can be permuted [6]. Exploring such symmetries in transformer-like architectures would indeed be interesting.
>
> To this end, we attempted to conduct an empirical study. However, to the best of our understanding, the open-source weight-matching (WM) implementation used by us does not handle these special symmetries by default. Operating on specific modules like attention requires significant code-level modifications due to the strict requirement for functional equivalence between the permuted and original models. We spent significant time attempting to extend this functionality. However, despite dedicated efforts, we fell short of time and could not obtain WM results with transformers during the discussion phase.
>
> To the best of our knowledge, prominent works [2, 3, 8, 9, 10] on loss landscape connectivity modulo permutations, largely consider only MLPs and CNNs, and exclude transformers. Although [6] mentions symmetries in attention modules (§3.3), they do not report experiments with transformers. Following this precedent, our experiments involving ResNets, DenseNets, VGGs, and two-layer MLPs sufficiently cover commonly used architectures in the literature. Extension to transformers and related architectures, while valuable, was beyond our primary scope. Given the absence of this analysis in prior work, it should ideally be the subject of a dedicated study on the star-domainness of transformer loss landscapes.
>
> **Possible future direction:** Recently, [7] exclusively studied transformer loss landscapes and introduced a new permutation algorithm, noting this problem as "non-trivial as they have more complicated connections than simpler MLP-like architectures" (§3). Their data-driven (in contrast with data-free WM) method focuses on text transformers. Adapting it to confirm the star domain conjecture for different transformer variants, including ViTs, could be promising.
>
> We sincerely thank the reviewer again for the suggestion. We hope that our response clarifies the challenges and scope of our work.
>
> We will address the first part of the reviewer's question (regarding theoretical verification) before the extended deadline of 23:59 on the 3rd of December, AoE.
>
> **References**
>
> [1-6] (As they appear in the reviewer's response above).
>
> [7] Verma, Neha., and Elbayad, Maha. "Merging text transformer models from different initializations." TMLR, November 2024.
>
> [8] Tatro, Norman, et al. "Optimizing mode connectivity via neuron alignment." NeurIPS 2022.
>
> [9] Peña, Fidel A. Guerrero, et al. "Re-basin via implicit sinkhorn differentiation." CVPR 2023.
>
> [10] Singh, Sidak Pal, and Martin Jaggi. "Model fusion via optimal transport." NeurIPS 2020.

---

> > ### Author Response · Authors · 2024-12-03
> > **Addressing the first part of Reviewer gkBc's question (regarding extension of Theorem 1)**
> >
> > We thank the reviewer for directing us to [1], which provides an explicit characterization of the geometry of the set of global minima for an overparameterized network. The work in [1] finds that this set forms a single manifold (a union of affine subspaces as pointed out by the reviewer), allowing any two global minima to be connected via piecewise linear paths within the manifold.
> >
> > We found this reference highly relevant to our work, as we are also focused on characterizing the geometry of the solution set. Extending Theorem 1 to the case of two-layer neural networks with nonlinear activations would indeed be interesting. However, several differences between the frameworks in [1] and our work prevent a direct application of their results to our findings:
> >
> > - While [1] explores the structure of the set of global minima by considering *all* permutations of a given solution, our approach introduces a "star model" and only considers *optimal* permutations that lead to linear connectivity with this star model.  More precisely, given a solution $\theta$, [1] considers all function-preserving permutations $\pi(\theta)$, $\pi \in S_n$ when describing the solution set. Instead, we find the optimal permutation $\pi_{\theta\rightarrow\theta^*}(\theta)$ that would linearly connect $\theta$ to the star model $\theta^*$. This distinction introduces challenges in directly leveraging the framework of [1] for our purposes.
> >
> > - [1] describes the global minimum manifold as the *union* of several affine subspaces. Potentially analogous to the star models in our work (albeit with the other differences we describe here) would be the *intersection* of these subspaces. Each affine subspace is, by definition, convex. If the intersection of all affine subspaces is non-empty, the minima inside this intersection would then be linearly connected to all other minima. Indeed, the authors of [1], in the proof of Theorem B.4, find a subset (say, $\widetilde{\Theta}$) of the set of global minima (say, $\Theta$) and show that any minimum in $\Theta \setminus \widetilde{\Theta}$ is connected to some minimum inside $\widetilde{\Theta}$ via a line segment. However, the connection to our work remains indirect as each minimum in $\Theta \setminus \widetilde{\Theta}$ can be connected to a different minimum in $\widetilde{\Theta}$.
> >
> > - Global minima in [1] are shown to be connected by *nonlinear* paths. In contrast, we show the existence of a "star model" and that all global minima are connected to this model via *linear* paths. Insights into how the simplicity of connecting paths evolves with increasing network width - such as, whether these paths simplify from nonlinear to linear as the width increases - would be valuable to our analysis. However, such aspects are not explicitly explored in [1].
> >
> > While [1] offers valuable insights, establishing an explicit connection to our framework and leveraging it to extend Theorem 1 remains non-trivial.
> >
> > Other references [2, 3, 4] mentioned by the reviewer are likewise highly relevant to our work, as we use the weight matching algorithm from [2] and our conjecture draws important parallels with the conjecture in [3]. [4] notably discovers the "variance collapse" problem in networks that contain batch normalization; we follow [2] and recalculate batch statistics to mitigate this problem.
> >
> > However, [2] and [4] do not explicitly provide theoretical exploration relevant to Theorem 1 in our work, and we have described the connection with [3] previously in our discussion (please also see our answer to Weakness 1 from Reviewer 3x6L). While we already cite [2, 3], we will make sure that [1, 4] are additionally cited.
> >
> > In conclusion, while all these works [1-4] are relevant to our paper, we have yet to find a strong connection that would enable us to extend Theorem 1 to nonlinear networks. We, therefore, leave this direction to future work. We may have missed another connection between [1-4] and our work that the reviewer had in mind. If this is the case, we would appreciate the reviewer letting us know.
> >
> > We remain grateful to the reviewer for spending time in proposing this direction and are always open to further feedback.
> >
> > ### References
> > [1] Simsek, Berfin, et al. "Geometry of the loss landscape in overparameterized neural networks: Symmetries and invariances." International Conference on Machine Learning. PMLR, 2021.
> >
> > [2] Ainsworth, Samuel K., Jonathan Hayase, and Siddhartha Srinivasa. "Git re-basin: Merging models modulo permutation symmetries." arXiv preprint arXiv:2209.04836 (2022).
> >
> > [3] Entezari, Rahim, et al. "The role of permutation invariance in linear mode connectivity of neural networks." arXiv preprint arXiv:2110.06296 (2021).
> >
> > [4] Jordan, Keller, et al. "Repair: Renormalizing permuted activations for interpolation repair." arXiv preprint arXiv:2211.08403 (2022).

---

### Official Review · Reviewer_3x6L · 2024-10-24

**Soundness:** 2
**Presentation:** 2
**Contribution:** 2
**Rating:** 6
**Confidence:** 3

**Summary:**

This paper conjectures that SGD solution set of neural networks is a star domain.
They have proved their conjecture for a toy model of neural networks, namely shallow linear neural networks with just one-dimensional input data . They also propose a Starlight algorithm to find a star model of a given task. It is mentioned that the benefit of having star domain for network solution is in Bayesian Model Averaging. Authors also supported their claims by proving empirical studies.

**Strengths:**

The paper is written clearly at some parts but I believe the writing can still be improved.
For example, it was not clear at the beginning of paper why such a star model property is important in NN, but authors explain this later
in their section 4.
The main contribution of the paper seems to be interesting, but I am still not convinced about their details.

**Weaknesses:**

I think beyond their Theorem 1 that is proved for a really toy model of neural networks under strict assumptions, there is no theoretical support over their claimed conjecture.
I also believe there are confusing points in the paper that need to be clarified, please see my question list below.

**Questions:**

1- This questions is related to the title and the whole paper `DO DEEP NEURAL NETWORK SOLUTIONS FORM A STAR DOMAIN?'
Do authors by 'solution' mean the global minima of objective function? If yes, how can authors make sure that global minima are reached in practice by SGD (objective functions are highly non-convex), let say in practical settings like limited width of the neural network? This question also concerns your empirical observations.
Also `The learning problem for a neural network is inherently characterized by a non-convex loss landscape, leading to multiple possible solutions rather than a singular one.' As far as I am aware, the first and main concern in optimization of NN is non-global optima solutions.
And could you please elaborate what is the role of SGD here? Why not other optimization methods?

2-Theorem 1 is targeting a toy model of neural networks (shallow, linear, ...), could you please elaborate how this theory under restrictive Assumption 1 can support your Conjecture? Could authors also elaborate on how Assumption 1 makes sense in practice?

3-Line 236: How can you reach all those solutions, while it is computationally too expensive? In fact, there are infinitely many solutions for neural networks, it kinda seems too far to extract a set of solutions. How large that set need to be to leave you a star point?

4-Line 267: It is not clear for me how authors handle finding optimal permutations? Could you please elaborate more?

5-Theorem 1 is valid for $m\to \infty$. How authors can handle this in practice?

6-Line 502, it is stated that this approach is useful to decrease computational expenses, but I am not sure how it can happened following your proposed setting in 236 and solving (3).

7-Better to clarify about importance of start shaped models sooner in text regarding your comments in section 4.2.

---

> ### Author Response · Authors · 2024-11-19
> **(1/3) Addressing Reviewer 3x6L's concerns: Weaknesses**
>
> Dear Reviewer 3x6L,
>
> Thank you for asking insightful questions. Below, we address your concerns.
>
> **Weakness 1: "I think beyond their Theorem 1 that is proved for a really toy model of neural networks under strict assumptions, there is no theoretical support over their claimed conjecture".**
>
> We acknowledge this concern. However, deriving theoretical results for full-fledged neural networks is notoriously difficult and, in many cases, infeasible due to their complexity. Theorem 1, proven for the simple yet abundantly challenging case of a 2-layer linear network represents our best attempt at providing a theoretical foundation to our findings.
>
> The most closely related work to our paper is the highly influential work by Entezari et al., who also tackled the simplified setup of a 2-layer neural network with a ReLU activation in between (Theorem 3.1 in their paper). However, Entezari et al. approach the problem differently. We highlight the differences below:
>
> | **Feature**                       | **Entezari et al.'s theory**               | **Our theory**                             |
> | --------------------------------- | ------------------------------------------ | ------------------------------------------ |
> | **Model type**                    | 2-layer NN with ReLU           | 2-layer linear NN              |
> | **Parameter set**                 | Randomly sampled parameters in a hypercube | Solutions derived from optimization        |
> | **Involves optimization problem** | No                                         | Yes                                        |
>
> By not considering actual solutions, Entezari et al.'s proof avoids the problem of characterising the solution set of a two-layer NN that contains a non-linearity between the layers. Our intention is not to critique their work but rather to highlight the broader challenges of developing theoretical proofs related to DNN solution sets. In our work, we take a different route by focusing on actual solutions of an optimization problem, although this necessitates dropping the non-linearity. We believe this puts the complexity of our theoretical results on a comparable footing with Entezari et al. while still offering a different perspective.
>
> To further address the gap in theoretical results, we provide extensive empirical evidence. We remain convinced that our experiments, covering three different architectures, three datasets including ImageNet, and ablation studies in Appendices C-D are extensive enough to constitute a sufficient contribution on their own. We hope that the current work serves as inspiration for other researchers to seek out more rigorous theoretical verification in future work.
>
> If you could kindly refer us to related works that you believe offer the kind of theoretical grounding you find compelling, we would be immensely grateful and would consider releasing additional theoretical results in future works.

---

> ### Author Response · Authors · 2024-11-19
> **(2/3) Addressing Reviewer 3x6L's concerns: Questions 1-2**
>
> **Question 1.1: "Do authors by 'solution' mean the global minima of objective function? If yes, how can authors make sure that global minima are reached in practice by SGD (objective functions are highly non-convex), let say in practical settings like limited width of the neural network? This question also concerns your empirical observations. Also The learning problem for a neural network is inherently characterized by a non-convex loss landscape, leading to multiple possible solutions rather than a singular one.' As far as I am aware, the first and main concern in optimization of NN is non-global optima solutions."**
>
> In general, 'solution' in our paper indeed refers to the global minima of the objective function.
>
> **In Theorem 1**, we ensure global minima by assuming precisely zero loss.
>
> **In Conjecture 2**, we also assume global minima that are theoretically reachable by SGD.
>
> **In practice**, it is tricky to determine whether global minima were reached. However we offer the following assurances.
>
> - For our **CIFAR10 and CIFAR100 experiments**, we reach extremely low training loss values (for example, training loss in the order of $10^{-3}$ for CIFAR10-ResNet18 models). These results strongly suggest that the obtained models are very close to the global minima of the objective function.
> - For our **ImageNet experiments**, the loss values are not exactly zero. However, the loss at the actual global minimum of the ImageNet classification task is an undetermined quantity because of inherent label noise. Nonetheless, our models reach state-of-the-art accuracy for the given dataset and architecture ($> 70\\%$ validation accuracy for ResNet18). Given this, we believe that our models serve as reasonable and sufficiently reliable proxies for global minima.
>
> Thank you again for raising this important question. We will include a concise clarification below Conjecture 2 and a more detailed one in the Appendix.
>
> **Question 1.2: "And could you please elaborate what is the role of SGD here? Why not other optimization methods?"**
>
> Our main study was performed with SGD solutions because literature on mode connectivity and solution set geometry has primarily focused on SGD solutions. Since our main contribution is not an extensive analysis of optimizer choices, but about examining the relaxation of the convexity conjecture in narrower networks, we have not extensively explored other optimizers. Nonetheless, we briefly explore Adam as another optimizer choice (lines 415-424 and Appendix D.2). We confirm that our conclusion remains the same under this ablation.
>
> **Question 2: "Theorem 1 is targeting a toy model of neural networks (shallow, linear, ...), could you please elaborate how this theory under restrictive Assumption 1 can support your Conjecture? Could authors also elaborate on how Assumption 1 makes sense in practice?"**
>
> We take both theoretical and empirical approaches to show that DNN solution sets are star domains.
> - From the theoretical perspective, we prove our result for a 2-layer linear NN. Assumption 1 is very restrictive indeed, but it is roughly at the same level of simplification performed elsewhere in the community (e.g. Theorem 3.1 in Entezari et al.). Generalising the theoretical results to more complex scenarios is difficult and should ideally be the subject of another paper; therefore, it is left as future work. Please also refer to our response to Weakness 1 above.
> - Complementing this, extensive analysis shows that our proposed conjecture aligns with experimental observations in more realistic settings, i.e., even when Assumption 1 does not hold.

---

> ### Author Response · Authors · 2024-11-19
> **(3/3) Addressing Reviewer 3x6L's concerns: Questions 3-7**
>
> **Question 3: "Line 236: How can you reach all those solutions, while it is computationally too expensive? In fact, there are infinitely many solutions for neural networks, it kinda seems too far to extract a set of solutions. How large that set need to be to leave you a star point?"**
>
> We are unsure if we understood this question correctly. May we rephrase it as "how can you train a star model by only sampling a few source models while the solution set contains infinitely many"?
>
> This is a great question. Our approach, which involves selecting $n$ source models and training a star model, can be viewed as using a subpopulation (the finite set of source models) to infer properties about the entire population (the infinite set of all possible solutions). Conceptually, this is similar to estimating a statistical property, like the mean of a random variable, by sampling from the corresponding distribution. Just as a subpopulation mean provides an estimate of the population mean, our finite sample of source models provides a basis for constructing a *candidate* star model that is expected to be connected to the rest of the "population", i.e., the infinite set of solutions, especially as our sample size $n$ used for training the star model increases. We show this effect in Figure 1, where "starness" increases as more and more source models are used. This approach enables us to form theories about the structure of the solution set without drawing infinitely many samples from the solution set.
>
> If you meant to ask a different question, we would be grateful for clarification.
>
> **Question 4: "Line 267: It is not clear for me how authors handle finding optimal permutations? Could you please elaborate more?"**
>
> As mentioned in the main text (lines 267-268 in the original PDF or lines 286-303 of the updated PDF), we use Ainsworth et al.'s Weight Matching (WM) method (detailed in §3.2 of their paper). Specifically, given the weight vectors $\theta_1$ and $\theta_2$ of two models respectively, this method seeks to solve for the permutation $\pi$ that maximizes the dot product of the weight vectors $\theta_1$ and $\pi(\theta_2)$.
>
> **Question 5: "Theorem 1 is valid for $m \rightarrow \infty$. How authors can handle this in practice?"**
>
> The $m \rightarrow \infty$ case in Theorem 1 signifies what happens as the width approaches infinity: our aim there is to validate the trend of an increasing proportion of star models in the solution set as the width $m$ increases. However, in practice, this implies that as network width increases (while still remaining finite), star-domainness becomes more and more likely to emerge. All of our experiments align with this idea, however, our results in Appendix A.2 (Table 3) especially show that in practice, star-domainness already appears at widths as low as $m=2$.
>
> **Question 6: "Line 502, it is stated that this approach is useful to decrease computational expenses, but I am not sure how it can happened following your proposed setting in 236 and solving (3)."**
>
> The advantage of lower computational expenses for our method compared to ensembling lies in *test-time complexity* rather than training-time complexity.
>
> We acknowledge that the Starlight algorithm incurs slightly higher computational cost than model ensembling during the training phase (note that in model ensembling, one still needs to train $n$ models, and the only complexity that our method adds is the final training run, where the star model is trained using Starlight). But while an ensembling approach requires storing $n$ models (memory complexity) and passing the input through $n$ different models (time complexity) *at test time*, our method requires only one model during inference. Effectively, any performance gains obtained using the star model are 'for free' in terms of time and memory complexity once the model has been deployed for inference.
>
> **Question 7: "Better to clarify about importance of start shaped models sooner in text regarding your comments in section 4.2."**
>
> Thank you for the suggestion. In addition to the relevant parts in the abstract (lines 020-023) and the introduction (lines 082-085), we will further highlight these points in §3.
>
> We look forward to your feedback.

---

> > ### Comment · Reviewer_3x6L · 2024-11-23
> >
> > I would like to thank the authors for their responses. Based on the provided information, the paper appears to rely more on empirical studies to support their conjecture rather than solid mathematical theories. I suggest that the authors elaborate further in their text on the technical challenges they encountered when extending their theory to practical scenarios and relaxing assumptions. This would be beneficial for future research in this direction. I would also be happy to raise my score to 6.

---

> > > ### Author Response · Authors · 2024-11-25
> > >
> > > We sincerely appreciate the reviewer's recognition of our contribution and fully agree with the suggestion to elaborate on the technical challenges we faced in our experiments.
> > >
> > > We will update our paper with this description before the discussion period closes. We will finalize this update after addressing Reviewer gkBc's question on extensions to other architectures, as it overlaps with some of the empirical challenges we plan to describe. We are conducting additional experiments to support this effort.
> > >
> > > We thank the reviewer once again for engaging with our work and helping us improve it further.
> > >
> > > Regards,
> > >
> > > Authors

---

### Official Review · Reviewer_eHk4 · 2024-11-03

**Soundness:** 2
**Presentation:** 3
**Contribution:** 2
**Rating:** 6
**Confidence:** 3

**Summary:**

#

The paper raises a conjecture if the trained neural networks, after taking into account for the permutation symmetries, form a star domain, i.e. there exists a neural network that is linearly connected to all the other networks.

Authors propose an algorithm to find the model that is star model by utilizing permutation matching algorithms and by defining a loss that minimizes the error barrier between the star model and other networks in the set.

Authors empirically validate the conjecture by showing that one can find a star model that is linearly connected to all the networks.

Authors show that model found by Starlight leads to mild improvement in generalization than individual models but still is a bit worse than model ensembling

**Strengths:**

- Paper is easy to follow and give sufficient background.
- Introduced the star-conjecture that is stronger than mode connectivity as in [1] but weaker than weak linear connectivity at defined in [2]

**Weaknesses:**

- Relationships with other notions of linear connectivity is not very well discussed.
- If I understand correctly, the experiments only conducted when there is weak linear connectivity modulo permutation holds since weak linear connectivity implies the star conjecture:
    - If weak linear connectivity modulo permutation holds, one can pick any model $\theta$ from the set $Z = \{\theta_1, \dots, \theta_N\}$ and find permutation $P_1, \dots, P_N$ that linearly connects the model $\theta_i$ to $theta$. This should lead to star connectivity
- Results for Starlight over model ensembling are not very extensive,

Minor comments:

- the bibliography needs many fixes. Specially because authors cite the arxiv version of the paper but should cite the publications

**Questions:**

I might be missing it but are there any experiments where Starlight algorithm finds a model where permutations fail to reduce the error barrier?

---

> ### Author Response · Authors · 2024-11-19
> **Addressing Reviewer eHk4's concerns.**
>
> Dear Reviewer eHk4,
>
> Thank you for your insightful assessment of our paper and for recognizing that it is easy to follow. Below, we address your concerns.
>
> **Weakness 1: "Relationships with other notions of linear connectivity is not very well discussed".**
>
> We agree that contextualizing the contribution in related literature is important. To this end, we discuss more than 25 papers in related work, including the seminal work by Entezari et al. We believe that the current amount of comparison against prior work is sufficient and self-contained enough to appropriately situate our conjecture against prior findings. However, we remain open to feedback. We would greatly appreciate it if you could kindly suggest any specific works that you believe would help make our discussion more thorough and rigorous.
>
> **Weakness 2: "The experiments only conducted when there is weak linear connectivity modulo permutation holds since weak linear connectivity implies the star conjecture".**
>
> Thank you for raising this concern. We believe there may have been a slight misunderstanding here, and would like to clarify. The star domain conjecture requires strictly less network width than linear connectivity. Even when linear connectivity *does not hold* due to insufficient network width, star-domainness can still be achieved. For example, please refer to Table 3, where in many instances, star-domainness holds while linear connectivity does not (all entries greater than $0.0$ and less than $1.0$ indicate cases where the solution set is not perfectly convex but contains at least one star model, i.e., it is a star domain).
>
> **Weakness 3: "Results for Starlight over model ensembling are not very extensive"**
>
> Model ensembling (§4.2) is not the main focus of our work. Our main thesis lies in showing that the shape of DNN solution sets is not necessarily convex as originally believed, and that the underlying geometry can be quite nuanced if one looks at narrower models. Specifically, the solution set slowly evolves from star-domainness to convexity as one increases network width. We believe that this observation adds meaningfully to existing notions of solution set geometry. We additionally show that our findings are potentially applicable in practice (through our results in §4). If there are any specific aspects where you find our results are not extensive, we would be more than happy to consider augmenting them. We look forward to your suggestions.
>
> **Weakness 4: "the bibliography needs many fixes. Specially because authors cite the arxiv version of the paper but should cite the publications"**.
>
> Thank you for your observation. We have now fixed all references to reflect the conference/journal publications wherever possible.
>
> **Question 1: "are there any experiments where Starlight algorithm finds a model where permutations fail to reduce the error barrier?"**
>
> We have so far not come across such failure cases among the settings that we tested (CIFAR and ImageNet on ResNet, DenseNet and VGG architectures). We would be happy to consider performing other experiments where you believe additional insights could be revealed.

---

> > ### Author Response · Authors · 2024-11-23
> > **Following up**
> >
> > Dear Reviewer eHk4,
> >
> > Since the discussion phase ends soon, we wanted to ensure that our response has adequately addressed your concerns. We would be happy to address any remaining concerns or questions.
> >
> > Thank you for reviewing our work,
> >
> > Authors

---

> ### Author Response · Authors · 2024-11-25
>
> Dear Reviewer eHk4,
>
> We would be grateful for an updated score if you have no further concerns. However, if you have any outstanding concerns, we would appreciate you sharing them now so that we have sufficient time to ensure the paper meets your expectations.
>
> Thank you again,
>
> Authors
>
> **Update**: We deeply appreciate the reviewer for recognizing our efforts and raising their score to 6. We remain open to any further feedback or suggestions.

---

### Official Review · Reviewer_H3sm · 2024-11-06

**Soundness:** 3
**Presentation:** 4
**Contribution:** 3
**Rating:** 6
**Confidence:** 4

**Summary:**

The paper proposes a weaker version of the conjecture that the minima recovered by SGD are convex up to permutations, proposing instead that this set is star-shaped up to permutations. A small theoretical motivation is given for shallow linear networks, but the main evidence is empirical, where the authors propose an algorithm to find the star model at the center of the start shape from a ensemble of trained networks. They then observe that the star-shaped conjecture is approximately true for networks where the convex conjecture is not.

Finally two alternatives are proposed to ensembling: sampling from the star shape, or taking the star model. Both seem to be promising.

**Strengths:**

The paper is well written, and the conjecture makes sense and is easy to understand. The experiments are quite convincing, though one could obviously alway try larger/deeper networks.

**Weaknesses:**

The authors do not propose any intuition why star-shape is the right choice to fix the convex up to permutation conjecture. In the theoretical results, can star-shapedness be guaranteed for significantly smaller widths than convexity?

Star-shape is significantly weaker than convexity (especially in high dimension, I would guess), so it is not completely surprising that one can reach a lower barrier. On the other hand switching from convexity to star shape makes this conjecture much harder to check and compute, since one has to find the star model at the center, and the experiments suggest that one needs a very large amount of models to find this star model accurately.

**Questions:**

See weaknesses.

---

> ### Author Response · Authors · 2024-11-19
> **Addressing Reviewer H3sm's concerns.**
>
> Dear Reviewer H3sm,
>
> Thank you for recognizing our work as well-written, and for finding our experiments convincing. Below, we address your concerns.
>
> **Strengths 1: "one could obviously alway try larger/deeper networks".**
>
> We resonate with this; we have made a concerted effort to incorporate a variety of network sizes, including very wide ones (e.g., Wide-ResNet $22\times8$) and fairly deep ones (ResNet-40). Depending on feasibility, we would be happy to consider testing any particular networks that you have in mind.
>
> **Weakness 1.1: "The authors do not propose any intuition why star-shape is the right choice to fix the convex up to permutation conjecture"**
>
> Thank you for raising this concern. We wish to clarify that our proposed star domain conjecture does not aim to "fix" the convexity conjecture. Instead, we intend to provide an additional understanding of the loss landscape *on top of* the convexity conjecture, which only states that given sufficient network width, the solution set is convex modulo permutations. The convexity conjecture alone does not provide intuition on how convexity is achieved as the width increases. We shed light on a transient stage between extremely thin models, where no connectivity is observed, and extremely wide models, where convexity holds. For the intermediate width case, our experiments reveal that the solution set is essentially a star domain and that the proportion of star models slowly increases until all models are star models, i.e., convexity has been attained. We believe that this intuition meaningfully contributes to the existing understanding of the geometry of solution sets for DNNs.
>
>
>
> **Weakness 1.2: "In the theoretical results, can star-shapedness be guaranteed for significantly smaller widths than convexity?"**
>
> Our theoretical results do not quantify the difference between the widths at which star-domainness and convexity appear, respectively. Theorem 1 claims that the proportion of star models in the solution set tends to 1 as width increases. To complement this, our empirical results answer the question of *how much smaller* the minimum network width required for star-domainness is, compared to the minimum network width required for convexity. For example, according to Table 3, when the tolerance for the barrier is $\epsilon=0.1$, the minimal required width for convexity is 32, while that for star-domainness is only 2.
>
>
> **Weakness 2.1: "Star-shape is significantly weaker than convexity (especially in high dimension, I would guess)"**
>
> This part is more nuanced when we take a detailed look. If higher parameter-space dimensionality implies a greater width of the network as well, then the maximal star-heldout barrier (related to the star domain conjecture) and the maximal heldout-heldout barrier (related to the convexity conjecture) will both tend to 0 as dimensionality increases. This implies that the star domain conjecture does not necessarily become weaker than the convexity conjecture as the dimensionality increases. On the contrary, the star domain conjecture slowly starts to *resemble* the convexity conjecture and the gap in weaknesses of the two conjectures *decreases*.
>
> **Weakness 2.2: " it is not completely surprising that one can reach a lower barrier."**
>
> The surprising part is the emergence of two classes of solutions: star models (connected to all other solutions) and non-star models (general solutions that are typically not connected to other solutions). We show that empirically, for any architecture of our choice, we could have obtained special star models that are connected to other independently trained models. This is in contrast with the convexity conjecture, which claims that solutions are all largely similar to each other in terms of connectivity.
>
>
> **Weakness 2.3: "Switching from convexity to star shape makes this conjecture much harder to check and compute, since one has to find the star model at the center, and the experiments suggest that one needs a very large amount of models to find this star model accurately."**
>
> While this is true, the thesis of our work is not concerned with efficiency in terms of compute or memory. We focus on the empirical verification of the existence of star models, in an effort to understand the loss landscape better. We show that our study holds novel practical implications for Bayesian model averaging and ensembling (as mentioned in your review and elaborated in §4 of our paper). Thus, our findings add an important layer of understanding on top of the existing convexity conjecture, and their significance is independent of the amount of compute used for obtaining them. However, we would be delighted to engage in further discussion about ideas to make our approach more efficient. We thank you again for raising this point.

---

> > ### Author Response · Authors · 2024-11-23
> > **Following up**
> >
> > Dear Reviewer H3sm,
> >
> > Since the discussion phase ends soon, we wanted to ensure that our response has adequately addressed your concerns. We would be happy to address any remaining concerns or questions.
> >
> > Thank you for reviewing our work,
> >
> > Authors

---

> > > ### Author Response · Authors · 2024-11-25
> > >
> > > Dear Reviewer H3sm,
> > >
> > > We wanted to send a brief reminder that we have addressed your concerns in detail in our first response. Further feedback from you is always welcome.
> > >
> > > Best,
> > >
> > > Authors

---

### Author Response · Authors · 2024-11-19

Dear reviewers,

Thank you for reading our paper and providing high-quality reviews. We will now begin posting responses. Some questions may take us longer to answer as we wish to make sure that we fully address your concerns. So, we will respond to those specific reviews later. We appreciate your understanding in this regard. We look forward to an interesting and productive discussion phase.

Best,

Authors

---

### Author Response · Authors · 2024-11-22
**Global response to all reviews: expressing gratitude and describing updates**

We sincerely thank the reviewers again for their efforts. We are delighted that our work was generally well-received.

The reviewers especially noted the breadth of our experimental results, recognizing them as "quite convincing" (H3sm) and stating that "the paper includes thorough empirical validation across various architectures and datasets, strengthening the claims made" (gkBc).

Reviewers also appreciated the paper's clarity, stating that it is "well written, and the conjecture makes sense and is easy to understand" (H3sm), and "easy to follow" (eHk4).

Some questions -- largely, requests for clarifications -- were raised. We especially appreciate the highly detailed questions from Reviewer 3x6L, which provided us with an opportunity to further clarify key points in our work. We have addressed all questions in the individual responses.

We have also uploaded a revised version of our submission PDF, with the following major changes:
- **Appendix A.2.3:** Added results for two-layer non-linear networks (gkBc).
- **Appendix B.4:** Clarified the nature of the solutions considered (3x6L).
- Replaced arXiv references with the corresponding journal or conference references wherever possible (eHk4).

We hope to have addressed the reviewers' concerns and are now looking forward to their responses. We are happy to answer further questions at any time.

---

### Author Response · Authors · 2024-12-03
**Closing remarks from authors**

As the discussion phase concludes, we appreciate the reviewers for unanimously accepting our paper, and the ACs for facilitating the discussion.

As noted in our previous global response, our updated submission PDF incorporates insightful suggestions from Reviewers eHk4, 3x6L, and gkBc. We also added Appendix A.3, describing potential directions for extending both our theoretical and practical results. Following Reviewer 3x6L's suggestion, this appendix also details the technical challenges involved in relaxing Assumption 1.

Thank you for your time and consideration!

Best,

Authors

---

### Meta-Review · Area_Chair_7f4a · 2024-12-23

**Metareview:**

This paper proposes and studies the “star domain conjecture,” which considers the shape of the global solutions of neural networks (up to permutations). The authors conjecture that there exist certain “star points” in the solution set such that the line segment between any element in the solution set and a star point entirely lies in the solution set. The conjecture is a stronger one than mode connectivity and a weaker one than the convexity (up to permutations) conjecture. The authors propose an algorithm named Starlight, aimed at finding a star solution given many “source solutions.” Using the algorithm, the paper verifies the conjecture by extensive experiments on different datasets, models, optimizers, etc. Lastly, the paper draws connections to Bayesian model averaging and model fusion, suggesting that the star solution can potentially be useful as a “summary model” of multiple different models.

The paper presents a bold and interesting conjecture and offers empirical/theoretical support of the claim. Numerical evaluations are quite thorough, in support of the proposed conjecture. The new conjecture sheds light on our understanding of the neural loss landscape.

The main weakness of the paper, pointed out by the reviewers, is that the paper does not provide a good insight or theory on why this star domain conjecture should be true in neural networks. The paper does have a theorem on a 2-layer linear neural network with univariate inputs, but the theorem does not quantify the effect of finite width. However, given the complex nature of the nonconvex training losses, it is deemed that the lack of formal theoretical explanations does not significantly harm the merit of the paper.

Overall, I believe that the paper can contribute to a deeper understanding of neural network training. I hence recommend acceptance.

**Additional Comments On Reviewer Discussion:**

Other comments from the reviewers were mostly clarification questions.

Reviewer gkBc asked about possible extensions of the theory to nonlinear networks and experiments on more modern architectures such as Transformers, along with concrete references. The authors diligently responded to the suggestions in the short time period. It is unfortunate that the authors were not able to make weight matching work for Transformers; hopefully some small-scale Transformer experiments can be added in the camera-ready version?

Also, as Reviewer H3sm suggested, it would be nice if one can prove that star convexity arises at a much smaller width than convexity, but this could be left for future work.

Lastly, the keyword star convexity reminded me (AC) of this work: https://arxiv.org/abs/1901.00451. I think the authors may find this paper relevant.

---

### Decision · Program_Chairs · 2025-01-22

Accept (Poster)